# Disentangling Images with Lie Group Transformations and Sparse Coding

**Ho Yin Chau**[1,5]                                          HCHAU630@BERKELEY.EDU
**Frank Qiu**[1,2,4]                                          FRANKQIU@BERKELEY.EDU
**Yubei Chen**[1,2,6]                                          YUBEIC@BERKELEY.EDU
**Bruno Olshausen**[1,2,3]                                  BAOLSHAUSEN@BERKELEY.EDU
[1]*Redwood Center,* [2]*BAIR,* [3]*Helen Wills Neuroscience Inst.,* [4]*Statistics Dept., UC Berkeley*
[5]*Center for Theoretical Neuroscience, Columbia University*
[6]*Center for Data Science, New York University*

**Editors:** Sophia Sanborn, Christian Shewmake, Simone Azeglio, Arianna Di Bernardo, Nina Miolane

## Abstract

Discrete spatial patterns and their continuous transformations are two important regularities in natural signals. Lie groups and representation theory are mathematical tools used in previous works to model continuous image transformations. On the other hand, sparse coding is an essential tool for learning dictionaries of discrete natural signal patterns. This paper combines these ideas in a Bayesian generative model that learns to disentangle spatial patterns and their continuous transformations in a completely unsupervised manner. Images are modeled as a sparse superposition of shape components followed by a transformation parameterized by $n$ continuous variables. The shape components and transformations are not predefined but are instead adapted to learn the data's symmetries. The constraint is that the transformations form a representation of an $n$-dimensional torus. Training the model on a dataset consisting of controlled geometric transformations of specific MNIST digits shows that it can recover these transformations along with the digits. Training on the full MNIST dataset shows that it can learn the basic digit shapes and the natural transformations such as shearing and stretching contained in this data. This work provides the simplest known Bayesian mathematical model for building unsupervised factorized representations.

**Keywords:** Sparse Coding, Lie Group Transformations, Disentanglement, Form and Motion

## 1. Introduction

A major challenge for both models of perception and unsupervised learning is to form disentangled representations of image data, in which variations along the latent dimensions explicitly reflect factors of variation in the visual world. An important dichotomy among these factors of variation is the distinction between discrete patterns vs. continuous transformations (Mumford and Desolneux, 2010), the former referring to factors such as local shape features or objects and the latter typically referring to geometric transformations such as translation, scaling, and rotation. Importantly, these factors are not overtly measurable but rather *entangled* in the pixel values of an image. Learning to disentangle the shapes and transformations inherent in image data is an important task that many previous works have attempted to address using architectures such as bilinear models, manifold models, and modified VAEs (Tenenbaum and Freeman, 2000; Grimes and Rao, 2005; Olshausen et al.,

2007; DiCarlo and Cox, 2007; Cadieu and Olshausen, 2011; Bengio et al., 2013; Cheung et al., 2014; Dupont, 2018).

One class of previous approaches has used Lie groups to model image transformations to varying degrees of generality (Rao and Ruderman, 1999; Miao and Rao, 2007; Culpepper and Olshausen, 2009; Sohl-Dickstein et al., 2010; Cohen and Welling, 2014, 2015; Gklezakos and Rao, 2017). A Lie group can be thought of as a parametric family of continuous transformations, and is a natural tool not only for modelling image transformations but also for learning disentangled representations more generally (Higgins et al., 2018; Pfau et al., 2020). However, all of these previous approaches focus solely on learning transformations but not the discrete patterns in the dataset. On the other hand, sparse coding is a widely known unsupervised algorithm for learning a dictionary of discrete spatial patterns from which images are composed (Olshausen and Field, 1997). Neurons in their hidden layer have been shown to recapitulate receptive field properties of V1 neurons after training on natural images, suggesting that the early human visual system may be employing the computational strategies of sparse coding. However, sparse coding does not model image transformations explicitly, and as a result, information about both shape and transformations are entangled in its representations.

We propose a novel unsupervised algorithm, Lie Group Sparse Coding (LSC), that combines the advantages of sparse coding and Lie group learning. Particularly, our work is inspired by the work of Cohen and Welling (2014), which introduces the mathematical framework of representation theory to disentanglement learning. We build on this framework by including a latent representation of shape via sparse coding, which is inspired by the work of Gklezakos and Rao (2017). The proposed algorithm infers both the sparse representation of shape and its transformation from an image. It learns the shape dictionary and transformation operators from the data through an iterative process akin to expectation-maximization. We demonstrate our model's capacity to disentangle representations of shape and transformation from controlled datasets where the ground truth is known and from the full MNIST dataset where both the underlying shape categories and factors of variation due to style are unknown and must be learned from the data. This work provides the simplest known Bayesian mathematical model for building unsupervised factorized representations.

## 2. Preliminaries

Sparse coding seeks to learn a dictionary of templates $\{\Phi_i\}$, such that each image $\mathbf{I}$ can be described by a sparse linear combination of these templates $\mathbf{I} = \sum_i \Phi_i \, \alpha_i$. However, as mentioned in the introduction, sparse coding does not model image transformations explicitly. Transforming an image changes its sparse code $\boldsymbol{\alpha} = [\alpha_1, \cdots, \alpha_K]^T$ in a non-equivariant manner, meaning form and transformations are entangled in the sparse code representation. To address this problem, we add an operator $\boldsymbol{T}(\mathbf{s})$ that explicitly models the transformations, so that images are now represented as transformations of patterns generated from the sparse coding model, $\mathbf{I} = \boldsymbol{T}(\mathbf{s})\boldsymbol{\Phi}\boldsymbol{\alpha}$, where $\boldsymbol{\Phi} = [\Phi_1, \cdots, \Phi_K]$.

Inspired by the work of Cohen and Welling (2014), we choose to model the transformations $\boldsymbol{T}(\mathbf{s})$ as actions of compact, connected, commutative (CCC) Lie groups on images. Many transformations - including rotations, rigid motion, and translations - can be understood as Lie groups, or more informally, a group of continuous symmetries of a space. CCC Lie

groups are a small subset of Lie groups that have a simple mathematical structure and are equivalent to $n$-dimensional tori (Dwyer and Wilkerson, 1998). By the Peter-Weyl theorem, these transformations can be decomposed as $\boldsymbol{T}(\mathbf{s}) = \boldsymbol{W}\boldsymbol{R}(\mathbf{s})\boldsymbol{W}^T$, where $\boldsymbol{W}$ is an orthogonal matrix, $\boldsymbol{\omega}_l \in \mathbb{Z}^n$, and $\boldsymbol{R}(\mathbf{s})$ is a block diagonal matrix containing 2D-rotation subblocks of the form

$$\boldsymbol{R}_l(\mathbf{s}) = \begin{bmatrix} \cos(\boldsymbol{\omega}_l^T\mathbf{s}) & -\sin(\boldsymbol{\omega}_l^T\mathbf{s}) \\ \sin(\boldsymbol{\omega}_l^T\mathbf{s}) & \cos(\boldsymbol{\omega}_l^T\mathbf{s}) \end{bmatrix}. \tag{1}$$

In practice, as we show in Section 4, this parametrization supports the learning of various common transformations such as translation, rotation, shearing, stretching, etc. The dependence on the parameter $\mathbf{s}$ takes a simple form of a block diagonal matrix with 2x2 blocks which allows for efficient inference and learning. Interested readers may refer to Appendix A for detailed explanations of the theory behind the parametrization $\boldsymbol{T}(\mathbf{s}) = \boldsymbol{W}\boldsymbol{R}(\mathbf{s})\boldsymbol{W}^T$. A minor downside of restricting our attention to CCC Lie groups is that it imposes several theoretical constraints on the class of learnable transformations: Compactness enforces the transformations as periodic; connectedness precludes discrete transformations like reflections; commutativity excludes non-commutative transformations such as 3D rotations. Fortunately, while many transformations violate these constraints in theory, we demonstrate in Section 4 that they can still be approximately learned in practice.

## 3. Algorithm

**3.1 Probabilistic Model.** Let $\mathbf{I} \in \mathbb{R}^D$ be the input image, where $D$ is even, and let $L \leq D/2$. We model $\mathbf{I}$ as

$$\mathbf{I} = \boldsymbol{W}\boldsymbol{R}(\mathbf{s})\boldsymbol{W}^T\boldsymbol{\Phi}\boldsymbol{\alpha} + \boldsymbol{\epsilon} \tag{2}$$

where $\boldsymbol{W} \in \mathbb{R}^{D \times 2L}$ is a matrix with orthonormal columns (i.e. $\boldsymbol{W}^T\boldsymbol{W} = \mathbb{1}$), $\boldsymbol{\Phi} \in \mathbb{R}^{D \times K}$ is the dictionary with each column having unit L2 norm, and $\boldsymbol{R}(\mathbf{s})$ is the block diagonal matrix with the subblocks $\boldsymbol{R}_l(\mathbf{s})$ defined in Equation (1), where $l = 1, \cdots, L$. The random variables in the model are $\mathbf{s}$, $\boldsymbol{\alpha}$, and $\boldsymbol{\epsilon}$, which are all independent of each other. The transformation parameter $\mathbf{s} \in \mathbb{R}^n$ is a random vector whose components $s_i$ are i.i.d. with $s_i \sim \text{Unif}(0, 2\pi)$. The sparse code $\boldsymbol{\alpha} \in \mathbb{R}^K$ is also a random vector whose components $\alpha_k$ are i.i.d. with $\alpha_k \sim \text{Exp}(\lambda)$, the exponential distribution. The random noise $\boldsymbol{\epsilon}$ is i.i.d. Gaussian with variance $\sigma^2$ and zero mean. Given an image $\mathbf{I}$, we infer the transformation parameters $\mathbf{s}$ and sparse code $\boldsymbol{\alpha}$ according to their posterior distribution. Given a large ensemble of images, we aim to learn the parameters $\boldsymbol{\theta} = \{\boldsymbol{W}, \boldsymbol{\Phi}\}$ by maximizing their log-likelihood. The inference and learning procedures are presented in 3.2.

The weights $\boldsymbol{\omega}_l \in \mathbb{Z}^n$, which appear in the block diagonal rotational matrix $\boldsymbol{R}(\mathbf{s})$, are chosen to evenly tile the $n$-dimensional domain, with the constraint that if $\boldsymbol{\omega}_l$ is chosen then $-\boldsymbol{\omega}_l$ is omitted. This is because, as shown in the derivation of $\boldsymbol{T}(\mathbf{s}) = \boldsymbol{W}\boldsymbol{R}(\mathbf{s})\boldsymbol{W}$ in Appendix C, each 2x2 block in $\boldsymbol{R}(\mathbf{s})$ is obtained by combining $\boldsymbol{\omega}_l$ terms with opposite signs in Equation (6). A multiplicity $m \geq 1$ is then assigned to the weights, meaning each $\boldsymbol{\omega}_l$ is repeated $m$ times. Finally, we select the first $L$ $\boldsymbol{\omega}_l$ sorted by ascending frequency $||\boldsymbol{\omega}_l||_2$.

This model combines aspects of sparse coding (Olshausen and Field, 1997) and the work by Cohen and Welling (2014) on learning irreducible representations of commutative Lie groups. If the term $\boldsymbol{W}\boldsymbol{R}(\mathbf{s})\boldsymbol{W}^T$ is replaced by the identity matrix, the model is identical to the sparse coding model of Olshausen and Field (1997). If $\boldsymbol{\Phi}\boldsymbol{\alpha}$ is replaced by a transformed

image $\mathbf{I}'$, and $n = 1$ (scalar $s$), the model is identical to the one-parameter transformation model by Cohen and Welling (2014).

---

**Algorithm 1:** Lie Group Sparse Coding Algorithm

---

$\boldsymbol{\theta} = \{\boldsymbol{W}, \boldsymbol{\Phi}\} \leftarrow \{\boldsymbol{W}_0, \boldsymbol{\Phi}_0\}$ ▷ Initialize model parameters
**while** $\boldsymbol{W}, \boldsymbol{\Phi}$ *not converged* **do**
    Get normalized image batch $\mathbf{I}$
    $\boldsymbol{\alpha} \leftarrow \boldsymbol{\alpha}_0$ ▷ Initialize sparse coefficients
    **for** $i \in \{1, \cdots, T\}$ **do** ▷ Compute $\hat{\boldsymbol{\alpha}} = \arg\max_{\boldsymbol{\alpha}} P_{\boldsymbol{\theta}}(\boldsymbol{\alpha}|\mathbf{I})$
        Compute $P_{\boldsymbol{\theta}}(\mathbf{s}|\mathbf{I}, \boldsymbol{\alpha})$
        $\Delta\boldsymbol{\alpha} \leftarrow \mathbb{E}_{\mathbf{s} \sim P_{\boldsymbol{\theta}}(\mathbf{s}|\mathbf{I}, \boldsymbol{\alpha})}[\nabla_{\boldsymbol{\alpha}} \ln P_{\boldsymbol{\theta}}(\mathbf{I}|\mathbf{s}, \boldsymbol{\alpha})] + \nabla_{\boldsymbol{\alpha}} \ln P_{\boldsymbol{\theta}}(\boldsymbol{\alpha})$ ▷ Compute gradient for $\boldsymbol{\alpha}$
        $\boldsymbol{\alpha} \leftarrow$ FISTA update$(\boldsymbol{\alpha}, \Delta\boldsymbol{\alpha})$ ▷ Update $\boldsymbol{\alpha}$ using FISTA
    **end**
    $\hat{\boldsymbol{\alpha}} \leftarrow \boldsymbol{\alpha}$
    Compute $P_{\boldsymbol{\theta}}(\mathbf{s}|\mathbf{I}, \hat{\boldsymbol{\alpha}})$
    $\Delta\boldsymbol{\Phi} \leftarrow \mathbb{E}_{\mathbf{s} \sim P_{\boldsymbol{\theta}}(\mathbf{s}|\mathbf{I}, \hat{\boldsymbol{\alpha}})}[\nabla_{\boldsymbol{\Phi}} \ln P_{\boldsymbol{\theta}}(\mathbf{I}|\mathbf{s}, \hat{\boldsymbol{\alpha}})]$ ▷ Compute approximate gradient for $\boldsymbol{\Phi}$
    $\boldsymbol{\Phi} \leftarrow$ Normalize$(\boldsymbol{\Phi} + \Delta\boldsymbol{\Phi})$ ▷ Update $\boldsymbol{\Phi}$ and normalize columns
    $\Delta\boldsymbol{W} \leftarrow \mathbb{E}_{\mathbf{s} \sim P_{\boldsymbol{\theta}}(\mathbf{s}|\mathbf{I}, \hat{\boldsymbol{\alpha}})}[\nabla_{\boldsymbol{W}} \ln P_{\boldsymbol{\theta}}(\mathbf{I}|\mathbf{s}, \hat{\boldsymbol{\alpha}})]$ ▷ Compute approximate gradient for $\boldsymbol{W}$
    $\boldsymbol{W} \leftarrow$ RiemannianAdam$(\boldsymbol{W}, \Delta\boldsymbol{W})$ ▷ Update $\boldsymbol{W}$ with RiemannianAdam optimizer
**end**

---

**3.2 Inference and Learning.** The inference and learning procedure is outlined in Algorithm 1. The general idea is as follows: to learn the model parameters $\boldsymbol{\theta} = \{\boldsymbol{W}, \boldsymbol{\Phi}\}$, we perform gradient ascent on their log-likelihood using the approximate gradient

$$\nabla_{\boldsymbol{\theta}} \ln P_{\boldsymbol{\theta}}(\mathbf{I}) \approx \mathbb{E}_{\mathbf{s} \sim P_{\boldsymbol{\theta}}(\mathbf{s}|\mathbf{I}, \hat{\boldsymbol{\alpha}})}[\nabla_{\boldsymbol{\theta}} \ln P_{\boldsymbol{\theta}}(\mathbf{I}|\mathbf{s}, \hat{\boldsymbol{\alpha}})] \tag{3}$$

where $\hat{\boldsymbol{\alpha}} = \arg\max_{\boldsymbol{\alpha}} P_{\boldsymbol{\theta}}(\boldsymbol{\alpha}|\mathbf{I})$ is the MAP estimate of the hidden variable $\boldsymbol{\alpha}$ (see Appendix E for derivation details). Notice moreover that

$$\nabla_{\boldsymbol{\theta}} \ln P_{\boldsymbol{\theta}}(\mathbf{I}|\hat{\boldsymbol{\alpha}}) = \mathbb{E}_{\mathbf{s} \sim P_{\boldsymbol{\theta}}(\mathbf{s}|\mathbf{I}, \hat{\boldsymbol{\alpha}})}[\nabla_{\boldsymbol{\theta}} \ln P_{\boldsymbol{\theta}}(\mathbf{I}|\mathbf{s}, \hat{\boldsymbol{\alpha}})] \tag{4}$$

and that the right hand side Equation (4) is just our approximate gradient in Equation (3). Hence, another interpretation is that we replaced the original objective function $\ln P_{\boldsymbol{\theta}}(\mathbf{I})$ by the approximate objective function $\ln P_{\boldsymbol{\theta}}(\mathbf{I}|\hat{\boldsymbol{\alpha}})$, which is much more computationally tractable than the original (see Appendix D for an explicit formula for $\ln P_{\boldsymbol{\theta}}(\mathbf{I}|\boldsymbol{\alpha})$, and Appendix E for derivation details). The approximation step assumes that the posterior distribution $P(\boldsymbol{\alpha}|\mathbf{I})$ is sharply peaked around $\hat{\boldsymbol{\alpha}}$, meaning $P(\boldsymbol{\alpha}|\mathbf{I}) \approx \delta(\boldsymbol{\alpha} - \hat{\boldsymbol{\alpha}})$, where $\delta(\mathbf{x})$ is the Dirac delta function. This is the same approximation used in the sparse coding algorithm by Olshausen and Field (1997). Also note the similarity between this approach and the EM algorithm, as each gradient step partially maximizes the expectation of the log likelihood with respect to the posterior distribution of the hidden variable given the current parameters. On the other hand, the MAP estimate $\hat{\boldsymbol{\alpha}}$ is computed by gradient ascent on the log-posterior, whose gradient is computed via

$$\nabla_{\boldsymbol{\alpha}} \ln P_{\boldsymbol{\theta}}(\boldsymbol{\alpha}|\mathbf{I}) = \mathbb{E}_{\mathbf{s} \sim P_{\boldsymbol{\theta}}(\mathbf{s}|\mathbf{I}, \boldsymbol{\alpha})}[\nabla_{\boldsymbol{\alpha}} \ln P_{\boldsymbol{\theta}}(\mathbf{I}|\mathbf{s}, \boldsymbol{\alpha})] + \nabla_{\boldsymbol{\alpha}} \ln P_{\boldsymbol{\theta}}(\boldsymbol{\alpha}) \tag{5}$$

(see Appendix E for derivation details). Note that both Equation (3) and Equation (5) require inferring the posterior distribution of the transformation variable $P_{\boldsymbol{\theta}}(\mathbf{s}|\mathbf{I}, \boldsymbol{\alpha})$. We show how this posterior distribution can be computed in Appendix D.

One may wonder whether the posterior distribution $P(\mathbf{s}|\mathbf{I}, \boldsymbol{\alpha})$ can be approximated by $\delta(\mathbf{s} - \hat{\mathbf{s}})$, just as we did for $\alpha$ in Equation (3). This would allow us to avoid computing the full distribution and use the point estimate $\hat{\mathbf{s}} = \arg\max_{\mathbf{s}} P_{\boldsymbol{\theta}}(\mathbf{s}|\mathbf{I}, \boldsymbol{\alpha})$ (optimized using gradient descent) in order to update the model parameters. We find empirically that using this approach leads to worse convergence of the model parameters. We believe there are two reasons for this: first, during the initial phase of training, the posterior distribution of $\mathbf{s}$ has many local extrema and hence using a single point estimate $\hat{\mathbf{s}}$ is a bad approximation; second, the presence of many local extrema during initial stages means it is easy to get stuck in a local minimum using gradient descent. Furthermore, when the number of transformation parameters is small (which is the case for the experiments in this paper), it is faster to compute the full distribution of $\mathbf{s}$ than performing gradient descent to find $\hat{\mathbf{s}}$, as the full distribution can be computed in a highly parallelized manner but gradient descent cannot.

Gradients computation involves the expectation term $\bar{\boldsymbol{R}} = \mathbb{E}_{\mathbf{s} \sim P_{\boldsymbol{\theta}}(\mathbf{s}|\mathbf{I}, \boldsymbol{\alpha})}[\boldsymbol{R}(\mathbf{s})]$ (see Appendix E for details), which is obtained by numerically integrating $\int_{\mathbf{s}} P_{\boldsymbol{\theta}}(\mathbf{s}|\mathbf{I}, \boldsymbol{\alpha})\boldsymbol{R}(\mathbf{s})$ with $N$ samples along each dimension. An efficient way of computing this quantity using Fast Fourier Transform is detailed in Cohen and Welling (2015), although numerical integration is adequate for our purposes. We used FISTA to greatly speed up the inference of $\boldsymbol{\alpha}$ by around a factor of 10 (Beck and Teboulle, 2009). While the objective $\ln P_{\boldsymbol{\theta}}(\boldsymbol{\alpha}|\mathbf{I})$ is not guaranteed to be convex in $\boldsymbol{\alpha}$, which violates one of the theoretical assumptions of FISTA, we find that in practice $\boldsymbol{\alpha}$ always converges well. Details of the usage of FISTA in LSC is provided in Appendix F.

As mentioned in section 3.1, there are two hard constraints on the model parameters $\boldsymbol{W}, \boldsymbol{\Phi}$: the columns of $\boldsymbol{W}$ must be orthonormal, and the columns of $\boldsymbol{\Phi}$ must have unit norms. The constraint on $\boldsymbol{W}$ comes from that we only want to learn orthogonal representations, while the unit norm constraint on $\boldsymbol{\Phi}$ prevents $\boldsymbol{\Phi}$ from growing without bound (see Olshausen and Field (1997)). We optimized $\boldsymbol{\Phi}$ by performing projected gradient descent, normalizing each column of $\boldsymbol{\Phi}$ after each gradient step. For $\boldsymbol{W}$, we used the Riemannian ADAM optimizer to optimize $\boldsymbol{W}$ on the Stiefel manifold (the manifold of matrices with orthogonal columns). We used the Riemmanian ADAM implementation from the python package geoopt (Kochurov et al., 2020). Using Riemannian ADAM instead of projected gradient descent, as is done in Cohen and Welling (2014), was empirically found to speed up convergence by around 3-4 times.

## 4. Experiments

To demonstrate that our algorithm can successfully disentangle different forms and transformation factors, we first train the model on two synthetic datasets in which the generative models are fully known. We set $K = 10$ and $n = 2$, meaning there are 10 dictionary templates and 2 latent dimensions for the transformation parameter $\mathbf{s}$. In the first dataset, we select one image from each of the 10 digit classes in 28x28 MNIST, then apply 6000 random 2D translations to each of the 10 selected images, totaling 60000 images. Both vertical and horizontal translations are drawn uniformly between $-7$ and 7 pixels. In the

second dataset, instead of 2D translations, we apply 6000 random rotations and scaling to the 10 images. Rotation is drawn uniformly between $-75°$ and $75°$, while scaling is drawn uniformly between 0.5 and 1.0. Figure 4 in Appendix G shows 80 images from each dataset.

For each dataset, LSC can learn the 10 digits and the two operators that generated it (training details in Appendix I). Figure 1 shows the learned $W$ matrices. The learned dictionary $\boldsymbol{\Phi}$ is shown within Figure 2. Notice that each of the learned dictionary template $\boldsymbol{\Phi}_i$ corresponds to one of the digits. Latent traversals of the two operators are shown at the bottom of Figure 2, in which we select 2 random images from the test set and apply the learned operator $\boldsymbol{T}(\mathbf{s})$ with varying $\mathbf{s}$ to those images. It is clear from the figure that the learned transformations are exactly the 2D translation operators and the rotation + scaling operators, respectively. Strikingly, even though the rotation + scaling dataset contains only rotations between $-75°$ and $75°$, the model learns the full $360°$ rotation. This ability to generalize and correctly extrapolate the transformation present in the dataset is a feature of the Lie group structure that is built into LSC.

One might notice a slight mixture of rotation and scaling in the latent traversal plots in Figure 2, which may seem to suggest that the algorithm failed to disentangle rotation and scaling completely. However, we note that there is no reason to require a one-to-one correspondence between $(s_1, s_2)$ and (rotation, scaling) since any other linear combination of these dimensions still allows these transformations to be performed perfectly.

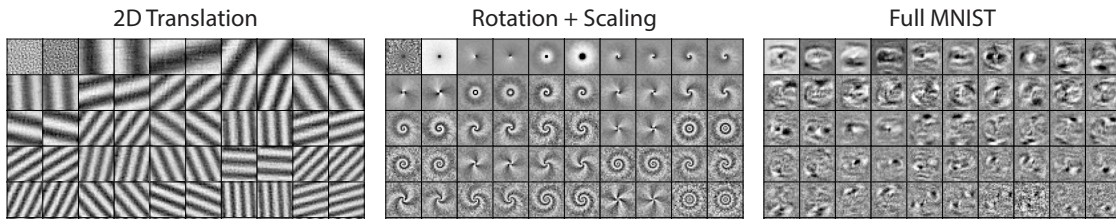

Figure 1: The first 50 columns of $\boldsymbol{W}$ learned on the three different datasets. Each image shows a column of $\boldsymbol{W}$, and are ordered by increasing values of $||\boldsymbol{\omega}||_2^2$

The inference process is demonstrated at the top of Figure 2. An image $\mathbf{I}$ is given to the model, which then performs the inference given in section 3.2 to yield the MAP estimate of the sparse coefficients $\hat{\boldsymbol{\alpha}}$ and the posterior distribution of the transformation parameter $P(\mathbf{s}|\mathbf{I}, \hat{\boldsymbol{\alpha}})$. A reconstruction of the input is then computed as $\hat{\mathbf{I}} = \boldsymbol{T}(\hat{\mathbf{s}})\boldsymbol{\Phi}\hat{\boldsymbol{\alpha}}$, where $\hat{\mathbf{s}} = \arg\max_{\mathbf{s}} P(\mathbf{s}|\mathbf{I}, \hat{\boldsymbol{\alpha}})$ is the MAP estimate of the $\mathbf{s}$. It can be seen from the figure that the inferred $\boldsymbol{\alpha}$ is 1-sparse. The posterior distribution of $\mathbf{s}$ is sharply peaked at a particular value.

We next trained our model on the full MNIST dataset to demonstrate its capacity to disentangle shape and transformations on data where the actual transformations are less clear and difficult to ascertain from first principles (see Appendix I for training details). The left side of Figure 3 are analogous to Figure 2. We see that LSC learns 9 out of the 10 digits with its dictionary $\boldsymbol{\Phi}$, while the latent traversal plots show that the model has learned horizontal stretching and shearing transformations. Figure 1 also shows the columns of the learned $W$ matrix. The reason that the dictionary did not learn the digit "1" is because, during inference, it always uses the learned horizontal stretching transform to "squeeze" a "0" into a "1", so that a separate "1" template is not necessary. In our experiments we found

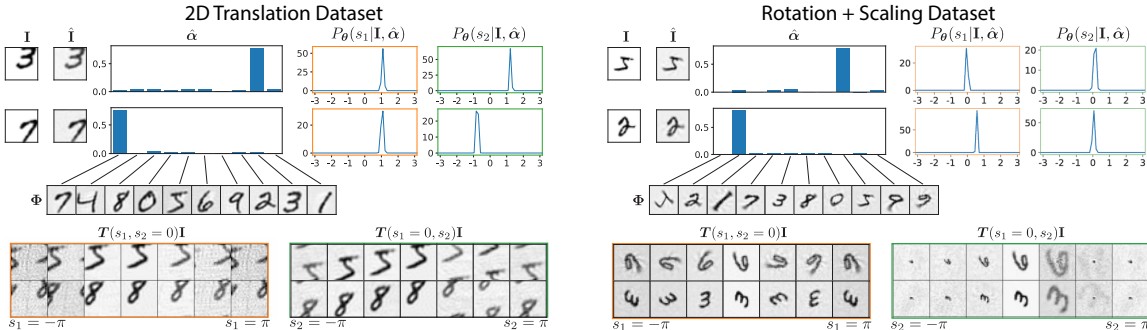

Figure 2: Top: Inference and image reconstruction for two example inputs $\mathbf{I}$ from each dataset. Bottom: Latent traversals of the transformation parameters $s_1$ and $s_2$, obtained by applying $\boldsymbol{T}(\mathbf{s})$ with varying values of $\mathbf{s}$ to two images from each test set. Orange figure shows latent traversal of $s_1$ from $-\pi$ to $\pi$, while green figure shows latent traversal of $s_2$ from $-\pi$ to $\pi$. The network has been trained on the respective datasets for 20 epochs. More examples are shown in Appendix H.

that one could learn the "1" template if a prior on $\mathbf{s}$ with a narrow peak near 0 is used instead of a uniform prior, with the intuition being that the narrow prior prevents large horizontal stretching transformations from being used to squeeze a "0" into a "1".

We compare LSC to sparse coding alone by training it on MNIST as well with the same number of dictionary elements. The learned dictionary $\boldsymbol{\Phi}$, as well as the inference process, is shown on the right of Figure 3. For comparison, the same two example inputs $\mathbf{I}$ were used in inference for both LSC and sparse coding. As can be seen, the reconstruction $\hat{\mathbf{I}}$ is significantly degraded without a transformation model. The inferred sparse representation $\hat{\boldsymbol{\alpha}}$ is also less sparse and less interpretable than LSC – e.g., the '8' is described as a combination of '2','0' and '9'. Also, note that the dictionary $\boldsymbol{\Phi}$ learned by sparse coding requires more than one template to capture the different poses of a digit, such as the slanted '1' in the 4th dictionary element and the upright '1' in the 7th element.

We demonstrate the improvement of LSC over sparse coding quantitatively in Table 1. We train both algorithms on the same datasets while ensuring that both models use the same dictionary size and sparsity cost. After training for 20 epochs, we evaluate the two algorithms on a test set and calculate the reconstructed images' SNR (signal-to-noise ratio). LSC outperforms sparse coding in all settings, and the improvement in SNR is over 10 times on the 2D translation dataset. This is particularly remarkable considering that we set the dimension of the transformation parameter $\mathbf{s}$ in LSC to be $n = 2$, meaning LSC only has two more degrees of freedom than sparse coding during the inference process.

## 5. Related Works

This work is inspired by Gklezakos and Rao (2017), who proposed the Transformational Sparse Coding algorithm (TSC) that combines ideas from Lie group theory with sparse coding like LSC. A significant difference is that in TSC, the group representation is predefined rather than learned; more concretely, the transformations in the generative model are fixed

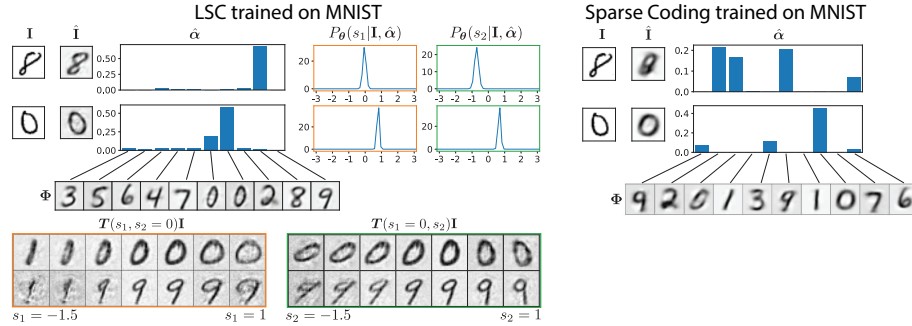

Figure 3: Top left: Inference and image reconstruction for two example inputs **I** from MNIST using LSC. Bottom left: Latent traversals of the transformation parameters $s_1$ and $s_2$ using LSC. Orange figure shows latent traversal of $s_1$ from $-1.5$ to 1, while green figure shows latent traversal of $s_2$ from $-1.5$ to 1. Top right: Inference and image reconstruction for the same two example inputs **I** using sparse coding. Both LSC and sparse coding have been trained on MNIST for 20 epochs. More examples are shown in Appendix H.

Table 1: Comparison of LSC and sparse coding using SNR (signal-to-noise ratio) of reconstructed images. Each row shows the mean and standard deviation of the SNR of reconstructed images ($\boldsymbol{T}(\hat{\mathbf{s}})\boldsymbol{\Phi}\hat{\boldsymbol{\alpha}}$ for LSC and $\boldsymbol{\Phi}\hat{\boldsymbol{\alpha}}$ for sparse coding) estimated from five trials in which we train both LSC and sparse coding on the same dataset with the same hyperparameters listed for 20 epochs (training details in Appendix I). Here $K$ is the number of dictionary templates and $\lambda$ is the sparsity cost.

| Dataset | Hyperparameters | LSC | Sparse Coding |
|---|---|---|---|
| 2D Translation Dataset | $K = 10, \lambda = 10.0$ | $\mathbf{31.0 \pm 1.0}$ | $2.21 \pm 0.01$ |
| Rotation + Scaling Dataset | $K = 10, \lambda = 10.0$ | $\mathbf{20.8 \pm 1.6}$ | $2.59 \pm 0.01$ |
| MNIST dataset | $K = 10, \lambda = 10.0$ | $\mathbf{4.6 \pm 0.2}$ | $2.99 \pm 0.01$ |
| MNIST dataset | $K = 100, \lambda = 1.0$ | $\mathbf{16.4 \pm 0.7}$ | $15.10 \pm 0.03$ |

to be the family of 2D affine transformations, whereas LSC allows for the learning of any transformations as long as they form a representation of a CCC Lie group. Another difference is that in TSC, images are modeled as being a combination of transformed variants of root templates. In contrast, in LSC, images are modeled as transformed versions of a combination of templates. In other words, TSC focuses on the transformations of local features, whereas LSC focuses on the global image transformations.

The inference and learning of transformations in our algorithm are based on the TSA algorithm (Toroidal Subgroup Analysis) by Cohen and Welling (2014) which learns the representation of a one-parameter subgroup of the maximal torus. There are two main differences: first, TSA learns image transformations given pairs of images with the same shape, whereas LSC learns representations of both shape and transformation without any prior knowledge of the shapes contained in the images; second, LSC generalizes the one-parameter subgroup representation learned in TSA to the representation of an *arbitrary*

$N$-dimensional torus, allowing for the learning of a wider variety of transformations. Cohen and Welling (2015) later extended their work to learning 3D object rotations using the group representation of $SO(3)$, although it again only learns the transformations but not the discrete spatial patterns.

LSC grew out of works on separating form and transformation using bilinear models. Bilinear models assume a generative process in which two vectors, one encoding form, and the other encoding transformation, combine multiplicatively, meaning the output is linear with respect to both vectors (Rao and Ballard, 1998; Tenenbaum and Freeman, 2000; Grimes and Rao, 2005; Olshausen et al., 2007; Memisevic and Hinton, 2010). Some bilinear models do not explicitly learn a transformation operator, but instead learn transformed versions of shape templates. As a result, the transformations learned will not easily generalize to new shapes (Tenenbaum and Freeman, 2000; Grimes and Rao, 2005). Other bilinear models do learn an explicit transformation operator, but because of the difficulty of inferring and learning large transformations, only local transformations are learned (Rao and Ballard, 1998; Olshausen et al., 2007). Attempts to learn larger transformations led to works on learning Lie group transformations. Typically, these transformations are learned from pairs of transformed images or a sequence/set of transforming images (Rao and Ruderman, 1999; Miao and Rao, 2007; Culpepper and Olshausen, 2009; Sohl-Dickstein et al., 2010). Early works learn the generator, or the Lie algebra, directly. Still, because of the computational intractability of gradient descent with respect to the matrix exponential, first-order Taylor expansion of the matrix exponential is used during learning, limiting the ability to learn from images with large transformations (Rao and Ruderman, 1999; Miao and Rao, 2007). An important innovation of Sohl-Dickstein et al. (2010) was to diagonalize the generator, which allows for tractable Lie group learning from large transformations. This idea was generalized by Cohen and Welling (2014) using representation theory, allowing for the learning of complex, large transformations such as 3D rotation (Cohen and Welling, 2015). This was an important generalization since describing the set of transformations on images as a representation of a Lie group rather than a Lie group itself, as is done in previous work, allows for the use of representation theory to simplify computations. For instance, the representation theory of the $N$-dimensional torus is used in deriving the simple parameterization of the transformation operator in this paper. We leave many related works that modify deep neural network architectures and objective functions to Appendix B.

## 6. Discussion

In this work, we study the problem of disentangling factors of variation in images, specifically discrete patterns vs. continuous transformations, which remains an open theoretical problem. To approach the problem, we combine Lie Group transformation learning and sparse coding within a Bayesian model. We show how spatial patterns and transformations can be learned as separate generative factors and inferred simultaneously. We wish to emphasize that the contribution is primarily in theory rather than a specific application. When the model is trained on synthetic datasets containing known geometric transformations, the digits are learned by the dictionary templates, and the applied transformations are learned by the transformation operator, providing a proof of concept demonstration of the feasibility of combining Lie Group transformation learning and sparse coding.

We wish to emphasize two main points about this work. Firstly, building on the foundational work of Cohen and Welling (2014), we show that incorporating the appropriate mathematical structure for describing transformations (Lie groups) enables a model to learn to disentangle shape and transformations in a network with a computationally simple structure. Although in principle a generic multilayer neural network could learn to approximate the inferential computations in this model, we conjecture that it would lead to a more complicated structure (when evaluated in terms of the number of weights and layers) and less robust performance in terms of its ability to generalize outside the training set. The representation theory of Lie groups leads us to a parameterization of the transformation operator that is both computationally efficient and effective at learning various transformations. Secondly, our Bayesian framework provides an advantage for learning a joint form and transformation model. While the usual approach is to jointly optimize the form and transformation parameters using gradient descent, a Bayesian approach reveals that it is better to integrate out the transformation parameters when optimizing the sparse coefficients. Empirically, we found that this approach achieves better convergence than the joint optimization, allowing the algorithm to reliably learn the correct transformations and shape dictionary. It is a promising future direction to use Resonator networks (Frady et al., 2020) for simultaneously factorizing $\boldsymbol{\alpha}$ and $\mathbf{s}$.

There are two main limitations to our model. 1) Our current way of computing the gradient of $\boldsymbol{\alpha}$ is not scalable to a large number of transformation parameters. It involves the expectation $\int_{\mathbf{s}} P_{\boldsymbol{\theta}}(\mathbf{s}|\mathbf{I}, \boldsymbol{\alpha}) \boldsymbol{R}(\mathbf{s})$, where the number of samples of $\mathbf{s}$ needed to compute the integral scales exponentially with $n$, or the dimension of $\mathbf{s}$. A potential direction to address this issue is to find a simple distribution approximating $P_{\boldsymbol{\theta}}(\mathbf{s}|\mathbf{I}, \boldsymbol{\alpha})$ that can be used for importance sampling. Another possibility is using MCMC methods. The second limitation to our model comes from the constraints on the transformations that can be learned, including orthogonality, compactness, connectedness, and commutativity. Though we showed in our experiments that some of these theoretical constraints are not quite problematic, there are still important transformations that cannot be learned as a result, such as global variations in contrast or a combination of rotation and translation. One possibility is to extend our current method and learn representations of a larger class of Lie groups, but it is unclear whether simple parameterizations exist for such groups. To address this issue, one possible future direction is to learn the representation of an arbitrary Lie group by learning the corresponding Lie algebra. Finally, the key idea of using a trainable Lie Group transformer module instead of a predefined transformation module like the spatial transformer (Jaderberg et al., 2015) may be highly useful for separating the transformation in deep neural networks. This is another interesting future direction.

## Acknowledgments

Research conducted by Yubei Chen at Redwood Center and BAIR was funded in part by NSF-IIS-1718991 and NSF-DGE-1106400. Bruno Olshausen's contributions were funded in part by NSF-IIS-1718991, NSF-DGE-1106400, and DARPA's Virtual Intellgence Processing program (SUPER-HD). Ho Yin Chau's contributions were funded in part by FAFSA. We thank Christian Shewmake for providing many helpful feedbacks during the preparation of this paper.

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

## Appendix A. The theory behind the parametrization

Many transformations - including rotations, rigid motion, and translations - can be understood as Lie groups, or more informally, a group of continuous symmetries of a space. Some examples include the groups of $n$-dimensional rotations $SO(n)$ and rigid motions $SE(n)$. When a Lie group deforms data, such as rotating an image, this constitutes the action of that Lie group $G$ on the vector space of data. If this action is linear, this amounts to a representation of $G$ - an instantiation of $G$ as a set of linear operators. More formally, a representation of $G$ on a vector space $V$ is a group homomorphism $\rho : G \to GL(V)$ which maps each element of $G$ to an invertible linear transformation on $V$.

In this paper, we consider the representation of compact, connected, commutative (CCC) Lie groups. We choose CCC Lie groups because they have straightforward representations. The Peter-Weyl theorem states that any unitary representation of a compact Lie group can be written as a direct sum of its irreducible representations; informally, a representation decomposes into its atomic parts - the irreducibles - which cannot be further decomposed. Since all irreducible representations of an $n$-dimensional torus (hence of CCC Lie groups, as they are equivalent) are of the form $\rho(e^{i\mathbf{s}}) = e^{i\boldsymbol{\omega}\cdot\mathbf{s}}$ for any $\boldsymbol{\omega} \in \mathbb{Z}^n$ (Kamnitzer, 2011), where $e^{i\mathbf{s}} = [e^{is_1}, e^{is_2}, \cdots, e^{is_n}]^T$ is a point on $\mathbb{T}^n$, any unitary representation a CCC Lie group is diagonal up to a unitary change of basis:

$$\rho(e^{i\boldsymbol{s}}) = \boldsymbol{V} \begin{bmatrix} e^{i\boldsymbol{\omega}_1^T \boldsymbol{s}} & & & \\ & e^{i\boldsymbol{\omega}_2^T \boldsymbol{s}} & & \\ & & \ddots & \\ & & & e^{i\boldsymbol{\omega}_D^T \boldsymbol{s}} \end{bmatrix} \boldsymbol{V}^H \equiv \boldsymbol{V} e^{\boldsymbol{\Sigma}(\mathbf{s})} \boldsymbol{V}^H \tag{6}$$

Since the data lives in the real vector space $\mathbb{R}^D$, we restrict $\rho$ to be real by choosing the $i\boldsymbol{\omega}_l$ in the diagonal matrix to come in conjugate pairs. After some simplification, this leads to the form $\boldsymbol{W}\boldsymbol{R}(\boldsymbol{s})\boldsymbol{W}^T$ (see Appendix C for details). In fact, *any* orthogonal representation of $\mathbb{T}^n$ can be written as $\boldsymbol{W}\boldsymbol{R}(\boldsymbol{s})\boldsymbol{W}^T$ (see Theorem 1 in Appendix C for details). In practice, we reduce the dimensionality of the model and improve efficiency by reducing the number of columns of $\boldsymbol{W}$ to $2L < D$.

A minor downside of restricting our attention to CCC Lie groups is that it imposes several theoretical constraints on the class of learnable transformations: Compactness enforces the transformations as periodic; connectedness precludes discrete transformations like reflections; commutativity excludes non-commutative transformations such as 3D rotations. Fortunately, while many transformations violate these constraints in theory, we demonstrate in Section 4 that they can still be approximately learned in practice.

## Appendix B. More related works from deep learning

A large body of work has appeared in recent years that modifies existing deep neural network architectures such as GAN and VAE to learn disentangled representations in an unsupervised manner (Cheung et al., 2014; Chen et al., 2016; Higgins et al., 2017; Dupont, 2018; Kim and Mnih, 2018; Chen et al., 2018). While these models are more general and potentially more powerful than LSC due to their many convolutional layers, they are also

substantially more complex (in terms of the number of layers) than LSC, which requires only one layer for transformation and one layer for sparse coding. The compactness of LSC is because we explicitly designed a layer to model Lie group transformations, which is the mathematical structure needed for the problems we are trying to solve. We speculate that a model with multiple such transformation layers could capture a broader range of image transformations than a generic multilayer convnet. There are also important works in supervised deep learning that include a specialized module to handle image transformation. Spatial Transformer Networks by Jaderberg et al. (2015) uses a differentiable transformer module that models affine transformations. Capsules use a group of neurons to collectively encode the probability of an object's presence and the pose/transformation of such an object (Hinton et al., 2011; Sabour et al., 2017; Hinton et al., 2018). Representation theory is used to develop new neural network layers that are equivariant to input transformations (Cohen and Welling, 2016, 2017; Cohen et al., 2018, 2019).

## Appendix C. Orthogonal representations of $\mathbb{T}^n$

**Theorem 1** *Any $D$-dimensional real, orthogonal representation of $\mathbb{T}^n$ can be written in the form $\boldsymbol{W}\boldsymbol{R}(\mathbf{s})\boldsymbol{W}^T$ where:*

1. *$\boldsymbol{R}(\mathbf{s})$ is a $D \times D$ block-diagonal matrix with $J+1$ blocks for some $J \leq D/2$*

2. *The first $J$ blocks of $\boldsymbol{R}(\mathbf{s})$ are of the form*

$$\begin{bmatrix} \cos(\boldsymbol{\omega}^T \boldsymbol{s}) & -\sin(\boldsymbol{\omega}^T \boldsymbol{s}) \\ \sin(\boldsymbol{\omega}^T \boldsymbol{s}) & \cos(\boldsymbol{\omega}^T \boldsymbol{s}) \end{bmatrix}$$

   *where $\boldsymbol{\omega} \in \mathbb{Z}^n$. We refer to this as a rotation block.*

3. *The last block of $\boldsymbol{R}(\mathbf{s})$ is a $(D-2J) \times (D-2J)$ identity matrix.*

4. *$\boldsymbol{W}$ is a $D \times D$ orthogonal matrix.*

*If $D$ is even, then $\boldsymbol{R}(\mathbf{s})$ is the block diagonal matrix with subblocks $\boldsymbol{R}_i(\mathbf{s})$ defined in Equation (1) in the main text, since in that case $D-2J$ is even, so the identity matrix block can be written as a direct sum of rotation blocks with $\boldsymbol{\omega} = 0$.*

**Proof**

Any unitary representation $\rho$ of $\mathbb{T}^n$ in $\mathbb{C}^D$ can be expressed in the form $\rho(e^{i\mathbf{s}}) = \boldsymbol{V}e^{\boldsymbol{\Sigma}(\mathbf{s})}\boldsymbol{V}^H$ (Equation (6) in the main text), where $\boldsymbol{\Sigma}(\mathbf{s})$ is a diagonal matrix with diagonal $(-i\boldsymbol{\omega}_1^T\mathbf{s}, \cdots, -i\boldsymbol{\omega}_D^T\mathbf{s})$.

We will show that for every $\boldsymbol{\omega}_j \neq 0$ there is a $\boldsymbol{\omega}_k$ such that $\boldsymbol{\omega}_k = -\boldsymbol{\omega}_j$. First, we assume that $\boldsymbol{\omega}_j \neq 0$ for all $j$. Our first step is to show that there exists some open ball $B \subset \mathbb{R}^n$ such that $e^{-i\boldsymbol{\omega}_j^T\mathbf{s}}$ have non-zero imaginary component for all $j$ and for all $\mathbf{s} \in B$. Consider the polynomial $f : U \subset \mathbb{R}^n \to \mathbb{R}$

$$f(\mathbf{s}) = \prod_j \boldsymbol{\omega}_j^T\mathbf{s}$$

where $U$ is a suitably small open subset of $\mathbb{R}^n$ such that $\boldsymbol{\omega}_j^T \mathbf{s} \in (-\pi, \pi)$ for all $j$. $f(\mathbf{s})$ is a non-zero function since $\boldsymbol{\omega}_j$'s are all non-zero by assumption, which implies there must be some open ball $B \subset U$ such that $f(\mathbf{s}) \neq 0$ for all $\mathbf{s} \in B$ (if not, then the zero set $f^{-1}(\{0\})$ is dense in $U$. But by continuity of $f$, $f^{-1}(\{0\})$ is a closed set, so $f^{-1}(\{0\}) = \overline{f^{-1}(\{0\})} = U$, meaning $f$ is a zero function, contradiction). This implies that for all $j$ and for all $\mathbf{s} \in B$, $\boldsymbol{\omega}_j^T \mathbf{s} \neq 0$, so $e^{-i\boldsymbol{\omega}_j^T \mathbf{s}} \neq 1$. Moreover, since $\boldsymbol{\omega}_j^T \mathbf{s} \in (-\pi, \pi)$, $e^{-i\boldsymbol{\omega}_j^T \mathbf{s}} \neq -1$. Since $e^{-i\boldsymbol{\omega}_j^T \mathbf{s}}$ lies on the unit circle but $e^{-i\boldsymbol{\omega}_j^T \mathbf{s}} \notin \{1, -1\}$, we conclude $e^{-i\boldsymbol{\omega}_j^T \mathbf{s}}$ must have non-zero imaginary component for all $j$ and for all $\mathbf{s} \in B$.

Now we show that for every $\boldsymbol{\omega}_j \neq 0$ there is a $\boldsymbol{\omega}_k$ such that $\boldsymbol{\omega}_k = -\boldsymbol{\omega}_j$. Consider the polynomial $g : B \subset \mathbb{R}^n \to \mathbb{R}$:

$$g(\mathbf{s}) = \prod_{j,k:j<k} (\boldsymbol{\omega}_j + \boldsymbol{\omega}_k)^T \mathbf{s}$$

where we continue to assume that $\boldsymbol{\omega}_j \neq 0$ for all $j$. By Lemma 3, every eigenvalue $e^{-i\boldsymbol{\omega}_j^T \mathbf{s}}$ of $\rho(e^{i\mathbf{s}})$ has a conjugate eigenvalue. Since $e^{-i\boldsymbol{\omega}_j^T \mathbf{s}}$ has non-zero imaginary part if $\mathbf{s} \in B$, for every $j$ there must be a $k \neq j$ such that $e^{-i\boldsymbol{\omega}_k^T \mathbf{s}} = e^{i\boldsymbol{\omega}_j^T \mathbf{s}}$ since $e^{-i\boldsymbol{\omega}_j^T \mathbf{s}}$ cannot be conjugate to itself. Therefore, for all $\mathbf{s} \in B$ at least one of the factors is 0, so $g(\mathbf{s}) = 0$ for all $\mathbf{s} \in B$. Lemma 4 implies that the $g(\mathbf{s})$ is the zero polynomial and hence $\boldsymbol{\omega}_{j^*} = -\boldsymbol{\omega}_{k^*}$ for some $j^*, k^*$ where $j^* \neq k^*$. Since conjugate eigenvalues have the same multiplicity by Lemma 3, we may remove the term $(\boldsymbol{\omega}_{j^*} + \boldsymbol{\omega}_{k^*})^T \mathbf{s}$ from $g(\mathbf{s})$ without changing the fact that $g(\mathbf{s}) = 0$. Then, we repeat the above procedure until all $\boldsymbol{\omega}$'s have been paired. Relaxing the assumption that $\boldsymbol{\omega}_j \neq 0$ for all $j$, we may apply the above argument by restricting our attention to only the set of non-zero $\boldsymbol{\omega}$'s. We conclude that for every $\boldsymbol{\omega}_j \neq 0$ there is a $k$ such that $\boldsymbol{\omega}_j = -\boldsymbol{\omega}_k$.

Therefore, the non-zero $\boldsymbol{\omega}$'s come in pairs $(\boldsymbol{\omega}, -\boldsymbol{\omega})$. If there are $J$ such pairs, WLOG we assume $\boldsymbol{\omega}_{2j} = -\boldsymbol{\omega}_{2j-1}$ for $1 \leq j \leq J$ and $\boldsymbol{\omega}_k = 0$ for $k > 2J$. Since the columns of $\boldsymbol{V}$ in $\rho(e^{i\mathbf{s}}) = \boldsymbol{V} e^{\boldsymbol{\Sigma}(\mathbf{s})} \boldsymbol{V}^H$ are the eigenvectors of $\rho(e^{i\mathbf{s}})$ and the eigenvalues $e^{i\boldsymbol{\omega}_j}$ come in conjugate pairs, Lemma 3 implies that WLOG we can assume the first $2J$ columns of $\boldsymbol{V}$ come in conjugate pairs. Moreover, we may also assume the last $D - 2J$ columns also come in conjugate pairs since eigenvectors of real eigenvalues also come in conjugate pairs by Lemma 3. Hence, WLOG $V_{2j} = \overline{V_{2j-1}}$ for all $j$.

Next, we show that the representation takes the form $\boldsymbol{W} \boldsymbol{R}(\mathbf{s}) \boldsymbol{W}^T$. Construct a block diagonal matrix $\boldsymbol{U}$ such that the first $J$ blocks are $2 \times 2$ blocks of the form

$$\frac{1}{\sqrt{2}} \begin{bmatrix} 1 & i \\ 1 & -i \end{bmatrix}$$

and the last block is just a $(D - 2J) \times (D - 2J)$ identity matrix. Using the fact that $\boldsymbol{U}$ is a unitary matrix, we can expand $\rho(e^{i\mathbf{s}})$ to get

$$\rho(e^{i\mathbf{s}}) = \boldsymbol{V} e^{\boldsymbol{\Sigma}(\mathbf{s})} \boldsymbol{V}^H = \boldsymbol{V} (\boldsymbol{U} \boldsymbol{U}^H) e^{\boldsymbol{\Sigma}(\mathbf{s})} (\boldsymbol{U} \boldsymbol{U}^H) \boldsymbol{V}^H = (\boldsymbol{V} \boldsymbol{U}) e^{\boldsymbol{U}^H \boldsymbol{\Sigma}(\mathbf{s}) \boldsymbol{U}} (\boldsymbol{V} \boldsymbol{U})^H$$

For the first $J$ blocks, we restrict our attention to a single $2 \times 2$ block of $\boldsymbol{U}^H \boldsymbol{\Sigma}(\mathbf{s}) \boldsymbol{U}$. Using $\boldsymbol{\omega}_{2j} = -\boldsymbol{\omega}_{2j-1}$, we simplify:

$$\left( \frac{1}{\sqrt{2}} \begin{bmatrix} 1 & 1 \\ -i & i \end{bmatrix} \right) \begin{bmatrix} i\boldsymbol{\omega}_{2j}^T \mathbf{s} & 0 \\ 0 & -i\boldsymbol{\omega}_{2j}^T \mathbf{s} \end{bmatrix} \left( \frac{1}{\sqrt{2}} \begin{bmatrix} 1 & i \\ 1 & -i \end{bmatrix} \right) = \begin{bmatrix} 0 & \boldsymbol{\omega}_{2j}^T \mathbf{s} \\ -\boldsymbol{\omega}_{2j}^T \mathbf{s} & 0 \end{bmatrix}$$

The exponential of this block is the $2 \times 2$ rotation matrix:

$$\exp\left(\begin{bmatrix} 0 & \boldsymbol{\omega}_{2j}^T \mathbf{s} \\ -\boldsymbol{\omega}_{2j}^T \mathbf{s} & 0 \end{bmatrix}\right) = \begin{bmatrix} \cos\left(\boldsymbol{\omega}_{2j}^T \mathbf{s}\right) & -\sin\left(\boldsymbol{\omega}_{2j}^T \mathbf{s}\right) \\ \sin\left(\boldsymbol{\omega}_{2j}^T \mathbf{s}\right) & \cos\left(\boldsymbol{\omega}_{2j}^T \mathbf{s}\right) \end{bmatrix}$$

and hence the first $J$ $2 \times 2$ blocks of $e^{\boldsymbol{U}^H \boldsymbol{\Sigma}(\mathbf{s})\boldsymbol{U}}$ are just $2 \times 2$ rotation blocks. On the other hand, the $(D-2J) \times (D-2J)$ block of $\boldsymbol{U}^H \Sigma(\mathbf{s})\boldsymbol{U}$ is a zero matrix, so the $(D-2J) \times (D-2J)$ block in $e^{\boldsymbol{U}^H \Sigma(\mathbf{s})\boldsymbol{U}}$ is the identity matrix. Thus, $e^{\boldsymbol{U}^H \boldsymbol{\Sigma}(\mathbf{s})\boldsymbol{U}} = \boldsymbol{R}(\mathbf{s})$.

Next we show that $\boldsymbol{VU}$ is a real orthogonal matrix and hence if we let $\boldsymbol{W} = \boldsymbol{VU}$ then $\rho(e^{i\mathbf{s}}) = \boldsymbol{WR}(\mathbf{s})\boldsymbol{W}^T$ as desired. Since $\boldsymbol{U}^H$ is block-diagonal, when computing the matrix product $\boldsymbol{U}^H \boldsymbol{V}^H$ we can individually consider each block of $\boldsymbol{U}^H$ multiplying its corresponding rows in $\boldsymbol{V}^H$. Recall that the columns of $\boldsymbol{V}$ come in conjugate pairs. Hence, restricting our attention to one pair of rows in $(\boldsymbol{VU})^H = \boldsymbol{U}^H \boldsymbol{V}^H$, we get

$$\left(\frac{1}{\sqrt{2}} \begin{bmatrix} 1 & 1 \\ -i & i \end{bmatrix}\right) \begin{bmatrix} \boldsymbol{v}^H \\ \overline{\boldsymbol{v}}^H \end{bmatrix} = \left(\frac{1}{\sqrt{2}} \begin{bmatrix} 2\mathrm{Re}(\boldsymbol{v}^H) \\ 2\mathrm{Im}(\boldsymbol{v}^H) \end{bmatrix}\right)$$

so all columns of $\boldsymbol{VU}$ are real. Because $\boldsymbol{VU}$ is a product of unitary matrices, it is also unitary, and $\boldsymbol{VU}$ must be real orthogonal Thus we have $\rho(e^{i\mathbf{s}}) = \boldsymbol{WR}(\mathbf{s})\boldsymbol{W}^T$.

■

**Remark 2** *The converse to the theorem is also true, namely that if $\rho$ is a map from $\mathbb{T}^n$ to $GL(D, \mathbb{R})$ such that $\rho(e^{i\mathbf{s}}) = \boldsymbol{WR}(\mathbf{s})\boldsymbol{W}^T$, then it is a $D$-dimensional real orthogonal representation of $\mathbb{T}^n$. This follows from the fact that $\boldsymbol{WR}(\mathbf{s})\boldsymbol{W}^T$ is orthogonal and $\boldsymbol{R}(\mathbf{s}_1)\boldsymbol{R}(\mathbf{s}_2) = \boldsymbol{R}(\mathbf{s}_1 + \mathbf{s}_2)$. Finally, we note that all orthogonal representations of $\mathbb{T}^n$ are actually special orthogonal representations of $\mathbb{T}^n$. This is also easy to see, since we know $\det(\rho(1)) = \det(\mathbb{1}) = 1$, so that if there exists some $e^{i\mathbf{s}} \in \mathbb{T}^n$ such that $\det\left(\rho(e^{i\mathbf{s}})\right) = -1$, then due to continuity of $\det$ and $\rho$ ($\rho$ is smooth by definition of representations of Lie groups), there must exist some $e^{i\mathbf{s}'}$ such that $\det\left(\rho(e^{i\mathbf{s}'})\right) = 0$, which is impossible.*

**Lemma 3** *If $A$ is a real matrix and $\lambda$ is a complex eigenvalue of $A$ with eigenvector $v$, then $\overline{\lambda}$ is also an eigenvalue with eigenvector $\overline{v}$. If $\lambda$ is an eigenvalue with multiplicity $n$, then $\overline{\lambda}$ is also an eigenvalue with multiplicity $n$.*

**Proof** Since the eigenvalues of $A$ are the roots of its characteristic polynomial $char(A) = \det(A - \lambda I)$, if $A$ is real then $char(A)$ is a real polynomial. Factorizing over $\mathbb{C}$:

$$char(A)(z) = c\, \Pi(z - r_i)$$

Suppose $r_1$ is a complex root with multiplicity $n > 1$. As $p(\bar{z}) = \overline{p(z)}$ for a real polynomial, $0 = char(A)(r_1) = char(A)(\bar{r_1})$, so $\overline{r_1}$ is also a root and hence an eigenvalue. Removing the factors $(z - r_1)(z - \overline{r_1})$, we repeat the same argument $n$ times to conclude that $\overline{r_1}$ is also an eigenvalue with multiplicity $n$.

If $\lambda$ is a complex eigenvalue of $A$ with eigenvector $v$, then $Av = \lambda v$. Taking the conjugate of both sides shows:

$$\overline{Av} = A\overline{v} = \overline{\lambda}\overline{v} = \overline{\lambda v}$$

∎

**Lemma 4** *For any* $p(\boldsymbol{x}) = p(x_1, \cdots, x_n) \in \mathbb{R}[X_1, \cdots, X_n]$, *if* $p(\boldsymbol{s}) = 0$ *for all* $\boldsymbol{s}$ *in an open set* $S$ *then* $p(\boldsymbol{x})$ *is the zero polynomial.*

**Proof** Let $\boldsymbol{t} \in S$. Then, $p'(\boldsymbol{x}) = p(\boldsymbol{x} - \boldsymbol{t}) = 0$ on some open $S'$ set containing $\boldsymbol{0}$. If two smooth functions coincide over some open set $U$, then their partial derivatives coincide on $U$. Hence, every partial derivative of $p'(\boldsymbol{x})$ is 0. As evaluating the partial derivatives of $p'(\boldsymbol{x})$ at $\boldsymbol{0}$ will return its coefficients, every coefficient is 0 and $p'(\boldsymbol{x})$ is the zero polynomial. As $p(\boldsymbol{x}) = p'(\boldsymbol{x} + t)$, $p(\boldsymbol{x})$ is also the zero polynomial. ∎

## Appendix D. Explicit formulae for $\ln P(I|\alpha)$ and $\ln P(s|I, \alpha)$

Although the main results presented in this paper assume a uniform prior on $\mathbf{s}$, a more general prior on $\mathbf{s}$ can be used, and we will derive the formula using this more general prior. This prior is actually the conjugate prior for our likelihood function, which is desirable as it gives a simple functional form for $\ln P(\mathbf{I}|\boldsymbol{\alpha})$. Using this more general prior is likely to be useful in problems where the true prior distribution of the transformation variables is known and can be well-approximated by this prior. Specifically, the general prior on $\mathbf{s}$ takes the form:

$$P(\mathbf{s}) = \frac{1}{Z(\boldsymbol{\eta})} \exp\Big(\sum_{l=1}^{L} \kappa_l \cos\big(\boldsymbol{\omega}_l^T \mathbf{s} - \mu_l\big)\Big) = \frac{1}{Z(\boldsymbol{\eta})} \exp\big(\boldsymbol{\eta}^T \boldsymbol{T}(\mathbf{s})\big)$$

which is a distribution from the exponential family with natural parameter:

$$\boldsymbol{\eta} = [\kappa_1 \cos(\mu_1), \kappa_1 \sin(\mu_1), \cdots, \kappa_L \cos(\mu_L), \kappa_L \sin(\mu_L)]^T$$
$$\equiv [\boldsymbol{\eta}_{11}, \boldsymbol{\eta}_{12}, \cdots, \boldsymbol{\eta}_{L1}, \boldsymbol{\eta}_{L2}]^T$$

sufficient statistics:

$$\boldsymbol{T}(\mathbf{s}) = [\cos\big(\boldsymbol{\omega}_1^T \mathbf{s}\big), \sin\big(\boldsymbol{\omega}_1^T \mathbf{s}\big), \cdots, \cos\big(\boldsymbol{\omega}_L^T \mathbf{s}\big), \sin\big(\boldsymbol{\omega}_L^T \mathbf{s}\big)]^T,$$

and normalization constant:

$$Z(\boldsymbol{\eta}) = \int_{\mathbf{s}} \exp\big(\boldsymbol{\eta}^T \boldsymbol{T}(\mathbf{s})\big) = \int_0^{2\pi} \cdots \int_0^{2\pi} \exp\big(\boldsymbol{\eta}^T \boldsymbol{T}(\mathbf{s})\big) d\mathbf{s}_1 \cdots d\mathbf{s}_n.$$

Note that one can recover the uniform prior on $\mathbf{s}$ by simply taking $\kappa_l = 0$ for all $l$. We also note that this is only a slightly more generalized version of the prior discovered by Cohen and Welling (2014), and that the following derivation of $\ln P(\mathbf{I}|\boldsymbol{\alpha})$ is also entirely due to Cohen and Welling (2014), with only slight modifications to adapt to our new model.

According to our model $\mathbf{I} = \boldsymbol{W}\boldsymbol{R}(\mathbf{s})\boldsymbol{W}^T\boldsymbol{\Phi}\boldsymbol{\alpha} + \boldsymbol{\epsilon}$, where $\boldsymbol{\epsilon} \sim \mathcal{N}(0, \sigma^2\mathbb{1})$, we have:

$$P(\mathbf{I}|\mathbf{s}, \boldsymbol{\alpha}) = \frac{1}{(2\pi\sigma^2)^{D/2}} \exp\left(-\frac{||\mathbf{I} - \boldsymbol{W}\boldsymbol{R}(\mathbf{s})\boldsymbol{W}^T\boldsymbol{\Phi}\boldsymbol{\alpha}||_2^2}{2\sigma^2}\right)$$

Now, for convenience, define $\boldsymbol{u} = \boldsymbol{W}\boldsymbol{\Phi}\boldsymbol{\alpha}$ and $\boldsymbol{v} = \boldsymbol{W}\mathbf{I}$. Moreover, define

$$\hat{\boldsymbol{\eta}} = [\hat{\eta}_{11}, \hat{\eta}_{12}, \hat{\eta}_{21}, \hat{\eta}_{22}, \cdots, \hat{\eta}_{L1}, \hat{\eta}_{L2}]$$

such that

$$\begin{bmatrix} \hat{\eta}_{l1} \\ \hat{\eta}_{l2} \end{bmatrix} = \begin{bmatrix} \eta_{l1} \\ \eta_{l2} \end{bmatrix} + \frac{1}{\sigma^2}\begin{bmatrix} u_{l1}v_{l1} + u_{l2}v_{l2} \\ u_{l1}v_{l2} - u_{l2}v_{l1} \end{bmatrix}$$

where

$$\begin{aligned} \boldsymbol{\eta} &= [\eta_{11}, \eta_{12}, \eta_{21}, \eta_{22}, \cdots, \eta_{L1}, \eta_{L2}] \\ \boldsymbol{u} &= [u_{11}, u_{12}, u_{21}, u_{22}, \cdots, u_{L1}, u_{L2}] \\ \boldsymbol{v} &= [v_{11}, v_{12}, v_{21}, v_{22}, \cdots, v_{L1}, v_{L2}] \end{aligned}$$

Then:

$$\begin{aligned} \ln P(\mathbf{I}|\boldsymbol{\alpha}) &= \ln \int_{\mathbf{s}} P(\mathbf{I}|\mathbf{s}, \boldsymbol{\alpha})P(\mathbf{s}) \\ &= \ln \int_{\mathbf{s}} \frac{1}{(2\pi\sigma^2)^{D/2}} \exp\left(-\frac{||\mathbf{I} - \boldsymbol{W}\boldsymbol{R}(\mathbf{s})\boldsymbol{W}^T\boldsymbol{\Phi}\boldsymbol{\alpha}||_2^2}{2\sigma^2}\right)\frac{1}{Z(\boldsymbol{\eta})}\exp(\boldsymbol{\eta}^T\boldsymbol{T}(\mathbf{s})) \\ &= \ln \int_{\mathbf{s}} \frac{1}{(2\pi\sigma^2)^{D/2}} \exp\left(-\frac{1}{2\sigma^2}(||\boldsymbol{W}^T\boldsymbol{\Phi}\boldsymbol{\alpha}||_2^2 + ||\mathbf{I}||_2^2) + \frac{1}{\sigma^2}\boldsymbol{v}^T\boldsymbol{R}(\mathbf{s})\boldsymbol{u}\right)\frac{1}{Z(\boldsymbol{\eta})}\exp(\boldsymbol{\eta}^T\boldsymbol{T}(\mathbf{s})) \\ &= \ln \left(\frac{\exp\left(-\frac{1}{2\sigma^2}(||\boldsymbol{W}^T\boldsymbol{\Phi}\boldsymbol{\alpha}||_2^2 + ||\mathbf{I}||_2^2)\right)}{(2\pi\sigma^2)^{D/2}}\frac{1}{Z(\boldsymbol{\eta})}\int_{\mathbf{s}}\exp\left(\boldsymbol{\eta}^T\boldsymbol{T}(\mathbf{s}) + \frac{1}{\sigma^2}\boldsymbol{v}^T\boldsymbol{R}(\mathbf{s})\boldsymbol{u}\right)\right) \\ &= -\frac{1}{2\sigma^2}(||\boldsymbol{W}^T\boldsymbol{\Phi}\boldsymbol{\alpha}||_2^2 + ||\mathbf{I}||_2^2) - \frac{D}{2}\ln(2\pi\sigma^2) - \ln Z(\boldsymbol{\eta}) + \ln\left(\int_{\mathbf{s}}\exp(\hat{\boldsymbol{\eta}}^T\boldsymbol{T}(\mathbf{s}))\right) \\ &= -\frac{1}{2\sigma^2}(||\boldsymbol{W}^T\boldsymbol{\Phi}\boldsymbol{\alpha}||_2^2 + ||\mathbf{I}||_2^2) - \frac{D}{2}\ln(2\pi\sigma^2) + \ln Z(\hat{\boldsymbol{\eta}}) - \ln Z(\boldsymbol{\eta}) \end{aligned}$$

where the integrands in step 4 and 5 are equal because

$$\begin{aligned} \boldsymbol{\eta}^T\boldsymbol{T}(\mathbf{s}) + \frac{1}{\sigma^2}\boldsymbol{v}^T\boldsymbol{R}(\mathbf{s})\boldsymbol{u} &= \sum_{l=1}^{L} \begin{bmatrix} \eta_{l1} \\ \eta_{l2} \end{bmatrix}^T \begin{bmatrix} \cos(\boldsymbol{\omega}_l^T\mathbf{s}) \\ \sin(\boldsymbol{\omega}_l^T\mathbf{s}) \end{bmatrix} + \frac{1}{\sigma^2}\begin{bmatrix} v_{l1} \\ v_{l2} \end{bmatrix}^T \begin{bmatrix} \cos(\boldsymbol{\omega}_l^T\mathbf{s}) & -\sin(\boldsymbol{\omega}_l^T\mathbf{s}) \\ \sin(\boldsymbol{\omega}_l^T\mathbf{s}) & \cos(\boldsymbol{\omega}_l^T\mathbf{s}) \end{bmatrix}\begin{bmatrix} u_{l1} \\ u_{l2} \end{bmatrix} \\ &= \sum_{l=1}^{L}\left(\begin{bmatrix} \eta_{l1} \\ \eta_{l2} \end{bmatrix} + \frac{1}{\sigma^2}\begin{bmatrix} u_{l1}v_{l1} + u_{l2}v_{l2} \\ u_{l1}v_{l2} - u_{l2}v_{l1} \end{bmatrix}\right)^T\begin{bmatrix} \cos(\boldsymbol{\omega}_l^T\mathbf{s}) \\ \sin(\boldsymbol{\omega}_l^T\mathbf{s}) \end{bmatrix} \\ &= \hat{\boldsymbol{\eta}}^T\boldsymbol{T}(\mathbf{s}) \end{aligned}$$

Note that the parameter $\hat{\boldsymbol{\eta}}$ determines the posterior distribution of $\mathbf{s}$, which is given by

$$P(\mathbf{s}|\mathbf{I}, \boldsymbol{\alpha}) = \frac{1}{Z(\hat{\boldsymbol{\eta}})}\exp(\hat{\boldsymbol{\eta}}^T\boldsymbol{T}(\mathbf{s})),$$

since $P(\mathbf{s}|\mathbf{I}, \boldsymbol{\alpha}) \propto P(\mathbf{I}|\mathbf{s}, \boldsymbol{\alpha})P(\mathbf{s}) \propto \exp(\hat{\boldsymbol{\eta}}^T\boldsymbol{T}(\mathbf{s}))$ (the last step can be seen from the derivation of $\ln P(\mathbf{I}|\boldsymbol{\alpha})$ where $P(\mathbf{I}|\mathbf{s}, \boldsymbol{\alpha})P(\mathbf{s})$ is the integrand). In order to compute $P(\mathbf{s}|\mathbf{I}, \boldsymbol{\alpha})$, we just compute $\hat{\boldsymbol{\eta}}$ and then apply the formula.

## Appendix E. Gradients of the model

In this section we provide details of the derivations for Equation (3), Equation (4), and Equation (5), as well as concrete expressions for the gradients. Notice that the derivations for Equation (3), Equation (4), and Equation (5) all employ the same "trick."

Equation (3):

$$
\begin{aligned}
\nabla_{\boldsymbol{\theta}} \ln P_{\boldsymbol{\theta}}(\mathbf{I}) &= \frac{1}{P_{\boldsymbol{\theta}}(\mathbf{I})} \nabla_{\boldsymbol{\theta}} P_{\boldsymbol{\theta}}(\mathbf{I}) \\
&= \frac{1}{P_{\boldsymbol{\theta}}(\mathbf{I})} \nabla_{\boldsymbol{\theta}} \int_{\mathbf{s}, \boldsymbol{\alpha}} P_{\boldsymbol{\theta}}(\mathbf{I}, \mathbf{s}, \boldsymbol{\alpha}) \\
&= \int_{\mathbf{s}, \boldsymbol{\alpha}} \frac{P_{\boldsymbol{\theta}}(\mathbf{I}, \mathbf{s}, \boldsymbol{\alpha})}{P_{\boldsymbol{\theta}}(\mathbf{I})} \nabla_{\boldsymbol{\theta}} \ln P_{\boldsymbol{\theta}}(\mathbf{I}, \mathbf{s}, \boldsymbol{\alpha}) \\
&= \int_{\mathbf{s}, \boldsymbol{\alpha}} P_{\boldsymbol{\theta}}(\mathbf{s}, \boldsymbol{\alpha}|\mathbf{I}) \nabla_{\boldsymbol{\theta}}(\ln P_{\boldsymbol{\theta}}(\mathbf{I}|\mathbf{s}, \boldsymbol{\alpha}) + \ln P(\mathbf{s}, \boldsymbol{\alpha})) \\
&= \int_{\boldsymbol{\alpha}} \left( P_{\boldsymbol{\theta}}(\boldsymbol{\alpha}|\mathbf{I}) \int_{\mathbf{s}} P_{\boldsymbol{\theta}}(\mathbf{s}|\mathbf{I}, \boldsymbol{\alpha}) \nabla_{\boldsymbol{\theta}} \ln P_{\boldsymbol{\theta}}(\mathbf{I}|\mathbf{s}, \boldsymbol{\alpha}) \right) \\
&\approx \int_{\boldsymbol{\alpha}} \left( \delta(\boldsymbol{\alpha} - \hat{\boldsymbol{\alpha}}) \int_{\mathbf{s}} P_{\boldsymbol{\theta}}(\mathbf{s}|\mathbf{I}, \boldsymbol{\alpha}) \nabla_{\boldsymbol{\theta}} \ln P_{\boldsymbol{\theta}}(\mathbf{I}|\mathbf{s}, \boldsymbol{\alpha}) \right) \\
&= \int_{\mathbf{s}} P_{\boldsymbol{\theta}}(\mathbf{s}|\mathbf{I}, \hat{\boldsymbol{\alpha}}) \nabla_{\boldsymbol{\theta}} \ln P_{\boldsymbol{\theta}}(\mathbf{I}|\mathbf{s}, \hat{\boldsymbol{\alpha}}) \\
&= \mathbb{E}_{\mathbf{s} \sim P_{\boldsymbol{\theta}}(\mathbf{s}|\mathbf{I}, \hat{\boldsymbol{\alpha}})}[\nabla_{\boldsymbol{\theta}} \ln P_{\boldsymbol{\theta}}(\mathbf{I}|\mathbf{s}, \hat{\boldsymbol{\alpha}})]
\end{aligned}
$$

Equation (4):

$$
\begin{aligned}
\nabla_{\boldsymbol{\theta}} \ln P_{\boldsymbol{\theta}}(\mathbf{I}|\hat{\boldsymbol{\alpha}}) &= \frac{1}{P_{\boldsymbol{\theta}}(\mathbf{I}|\hat{\boldsymbol{\alpha}})} \nabla_{\boldsymbol{\theta}} P_{\boldsymbol{\theta}}(\mathbf{I}|\hat{\boldsymbol{\alpha}}) \\
&= \frac{1}{P_{\boldsymbol{\theta}}(\mathbf{I}|\hat{\boldsymbol{\alpha}})} \nabla_{\boldsymbol{\theta}} \int_{\mathbf{s}} P_{\boldsymbol{\theta}}(\mathbf{I}, \mathbf{s}|\hat{\boldsymbol{\alpha}}) \\
&= \int_{\mathbf{s}} \frac{P_{\boldsymbol{\theta}}(\mathbf{I}, \mathbf{s}|\hat{\boldsymbol{\alpha}})}{P_{\boldsymbol{\theta}}(\mathbf{I}|\hat{\boldsymbol{\alpha}})} \nabla_{\boldsymbol{\theta}} \ln P_{\boldsymbol{\theta}}(\mathbf{I}, \mathbf{s}|\hat{\boldsymbol{\alpha}}) \\
&= \int_{\mathbf{s}} P_{\boldsymbol{\theta}}(\mathbf{s}|\mathbf{I}, \hat{\boldsymbol{\alpha}}) \nabla_{\boldsymbol{\theta}}(\ln P_{\boldsymbol{\theta}}(\mathbf{I}|\mathbf{s}, \hat{\boldsymbol{\alpha}}) + \ln P(\mathbf{s}|\hat{\boldsymbol{\alpha}})) \\
&= \int_{\mathbf{s}} P_{\boldsymbol{\theta}}(\mathbf{s}|\mathbf{I}, \hat{\boldsymbol{\alpha}}) \nabla_{\boldsymbol{\theta}} \ln P_{\boldsymbol{\theta}}(\mathbf{I}|\mathbf{s}, \hat{\boldsymbol{\alpha}}) \\
&= \mathbb{E}_{\mathbf{s} \sim P_{\boldsymbol{\theta}}(\mathbf{s}|\mathbf{I}, \hat{\boldsymbol{\alpha}})}[\nabla_{\boldsymbol{\theta}} \ln P_{\boldsymbol{\theta}}(\mathbf{I}|\mathbf{s}, \hat{\boldsymbol{\alpha}})]
\end{aligned}
$$

Equation (5):

$$
\begin{aligned}
\nabla_{\boldsymbol{\alpha}} \ln P_{\boldsymbol{\theta}}(\boldsymbol{\alpha}|\mathbf{I}) &= \nabla_{\boldsymbol{\alpha}} \ln P_{\boldsymbol{\theta}}(\mathbf{I}|\boldsymbol{\alpha}) + \nabla_{\boldsymbol{\alpha}} \ln P(\boldsymbol{\alpha}) \\
&= \mathbb{E}_{\mathbf{s} \sim P_{\boldsymbol{\theta}}(\mathbf{s}|\mathbf{I}, \boldsymbol{\alpha})}[\nabla_{\boldsymbol{\alpha}} \ln P_{\boldsymbol{\theta}}(\mathbf{I}|\mathbf{s}, \boldsymbol{\alpha})] + \nabla_{\boldsymbol{\alpha}} \ln P(\boldsymbol{\alpha})
\end{aligned}
$$

where in the last step we applied the Equation (4) with $\hat{\boldsymbol{\alpha}}$ replaced by $\boldsymbol{\alpha}$.

Next we provide concrete expressions for the various gradients:

$$\nabla_{\boldsymbol{\alpha}} \ln P_{\boldsymbol{\theta}}(\boldsymbol{\alpha}|\mathbf{I}) = \mathbb{E}_{\mathbf{s} \sim P_{\boldsymbol{\theta}}(\mathbf{s}|\mathbf{I},\boldsymbol{\alpha})}[\nabla_{\boldsymbol{\alpha}} \ln P_{\boldsymbol{\theta}}(\mathbf{I}|\mathbf{s},\boldsymbol{\alpha})] + \nabla_{\boldsymbol{\alpha}} \ln P(\boldsymbol{\alpha})$$

$$= \mathbb{E}_{\mathbf{s} \sim P_{\boldsymbol{\theta}}(\mathbf{s}|\mathbf{I},\boldsymbol{\alpha})}[\frac{1}{\sigma^2}\boldsymbol{\Phi}^T \boldsymbol{T}(\mathbf{s})^T \boldsymbol{\epsilon}] - \lambda \mathrm{sign}(\boldsymbol{\alpha})$$

$$= \frac{1}{\sigma^2}\boldsymbol{\Phi}^T \boldsymbol{W}(\bar{\boldsymbol{R}}^T \boldsymbol{W}^T \mathbf{I} - \boldsymbol{W}^T \boldsymbol{\Phi}\boldsymbol{\alpha}) - \lambda \mathrm{sign}(\boldsymbol{\alpha})$$

where $\boldsymbol{\epsilon} = \mathbf{I} - \boldsymbol{T}(\mathbf{s})\boldsymbol{\Phi}\boldsymbol{\alpha}$ and $\bar{\boldsymbol{R}} = \mathbb{E}_{\mathbf{s} \sim P_{\boldsymbol{\theta}}(\mathbf{s}|\mathbf{I},\boldsymbol{\alpha})}[\boldsymbol{R}(\mathbf{s})]$.

$$\nabla_{\boldsymbol{\Phi}} \ln P_{\boldsymbol{\theta}}(\mathbf{I}) \approx \mathbb{E}_{\mathbf{s} \sim P_{\boldsymbol{\theta}}(\mathbf{s}|\mathbf{I},\hat{\boldsymbol{\alpha}})}[\nabla_{\boldsymbol{\Phi}} \ln P_{\boldsymbol{\theta}}(\mathbf{I}|\mathbf{s},\hat{\boldsymbol{\alpha}})]$$

$$= \mathbb{E}_{\mathbf{s} \sim P_{\boldsymbol{\theta}}(\mathbf{s}|\mathbf{I},\hat{\boldsymbol{\alpha}})}[\frac{1}{\sigma^2}\boldsymbol{T}(\mathbf{s})^T \hat{\boldsymbol{\epsilon}}\hat{\boldsymbol{\alpha}}^T]$$

$$= \frac{1}{\sigma^2}\boldsymbol{W}(\bar{\boldsymbol{R}}^T \boldsymbol{W}^T \mathbf{I} - \boldsymbol{W}^T \boldsymbol{\Phi}\hat{\boldsymbol{\alpha}})\hat{\boldsymbol{\alpha}}^T$$

where $\hat{\boldsymbol{\epsilon}} = \mathbf{I} - \boldsymbol{T}(\mathbf{s})\boldsymbol{\Phi}\hat{\boldsymbol{\alpha}}$.

$$\nabla_{\boldsymbol{W}} \ln P_{\boldsymbol{\theta}}(\mathbf{I}) \approx \mathbb{E}_{\mathbf{s} \sim P_{\boldsymbol{\theta}}(\mathbf{s}|\mathbf{I},\hat{\boldsymbol{\alpha}})}[\nabla_{\boldsymbol{W}} \ln P_{\boldsymbol{\theta}}(\mathbf{I}|\mathbf{s},\hat{\boldsymbol{\alpha}})]$$

$$= \mathbb{E}_{\mathbf{s} \sim P_{\boldsymbol{\theta}}(\mathbf{s}|\mathbf{I},\hat{\boldsymbol{\alpha}})}[\frac{1}{\sigma^2}(\hat{\boldsymbol{\epsilon}}(\boldsymbol{T}(\mathbf{s})\boldsymbol{\Phi}\hat{\boldsymbol{\alpha}})^T + \boldsymbol{\Phi}\hat{\boldsymbol{\alpha}}(\boldsymbol{T}(\mathbf{s})^T \hat{\boldsymbol{\epsilon}})^T)\boldsymbol{W}]$$

$$= \frac{1}{\sigma^2}(\boldsymbol{\Phi}\hat{\boldsymbol{\alpha}}(\boldsymbol{W}^T \mathbf{I})^T \bar{\boldsymbol{R}} + \mathbf{I}(\boldsymbol{W}^T \boldsymbol{\Phi}\hat{\boldsymbol{\alpha}})^T \bar{\boldsymbol{R}}^T - \boldsymbol{\Phi}\hat{\boldsymbol{\alpha}}(\boldsymbol{W}^T \boldsymbol{\Phi}\hat{\boldsymbol{\alpha}})^T - \mathbb{E}_{\mathbf{s} \sim P_{\boldsymbol{\theta}}(\mathbf{s}|\mathbf{I},\hat{\boldsymbol{\alpha}})}[\hat{\mathbf{I}}\hat{\mathbf{I}}^T]\boldsymbol{W})$$

$$= \frac{1}{\sigma^2}(\boldsymbol{\Phi}\hat{\boldsymbol{\alpha}}(\bar{\boldsymbol{R}}^T \boldsymbol{W}^T \mathbf{I})^T + \mathbf{I}(\bar{\boldsymbol{R}}\boldsymbol{W}^T \boldsymbol{\Phi}\hat{\boldsymbol{\alpha}})^T - \boldsymbol{\Phi}\hat{\boldsymbol{\alpha}}(\boldsymbol{W}^T \boldsymbol{\Phi}\hat{\boldsymbol{\alpha}})^T$$

$$- \boldsymbol{W}\mathbb{E}_{\mathbf{s} \sim P_{\boldsymbol{\theta}}(\mathbf{s}|\mathbf{I},\hat{\boldsymbol{\alpha}})}[\boldsymbol{R}(\mathbf{s})\boldsymbol{W}^T \boldsymbol{\Phi}\hat{\boldsymbol{\alpha}}\hat{\boldsymbol{\alpha}}^T \boldsymbol{\Phi}^T \boldsymbol{W}\boldsymbol{R}(\mathbf{s})^T])$$

where $\hat{\mathbf{I}} = \boldsymbol{T}(\mathbf{s})\boldsymbol{\Phi}\hat{\boldsymbol{\alpha}}$

As seen, the gradient for $\boldsymbol{W}$ is quite difficult to compute. In our implementation, we used an approximation that results in a much simpler expression for the gradient for $\boldsymbol{W}$ derived above, by assuming independence between the term $\boldsymbol{T}(\mathbf{s})$ and $\hat{\boldsymbol{\epsilon}}$, which are both dependent on the random variable $\mathbf{s}$. For consistency, we also applied the same approximation to both $\boldsymbol{\alpha}$ and $\boldsymbol{\Phi}$ gradients. More explicitly, we use the following approximate gradients:

$$\nabla_{\boldsymbol{\alpha}} \ln P_{\boldsymbol{\theta}}(\boldsymbol{\alpha}|\mathbf{I}) = \mathbb{E}_{\mathbf{s} \sim P_{\boldsymbol{\theta}}(\mathbf{s}|\mathbf{I},\boldsymbol{\alpha})}[\frac{1}{\sigma^2}\boldsymbol{\Phi}^T \boldsymbol{T}(\mathbf{s})^T \boldsymbol{\epsilon}] - \lambda \mathrm{sign}(\boldsymbol{\alpha})$$

$$\approx \frac{1}{\sigma^2}\boldsymbol{\Phi}^T \bar{\boldsymbol{T}}^T \hat{\boldsymbol{\epsilon}} - \lambda \mathrm{sign}(\boldsymbol{\alpha})$$

$$\nabla_{\boldsymbol{\Phi}} \ln P_{\boldsymbol{\theta}}(\mathbf{I}) \approx \mathbb{E}_{\mathbf{s} \sim P_{\boldsymbol{\theta}}(\mathbf{s}|\mathbf{I},\hat{\boldsymbol{\alpha}})}[\frac{1}{\sigma^2}\boldsymbol{T}(\mathbf{s})^T \hat{\boldsymbol{\epsilon}}\hat{\boldsymbol{\alpha}}^T]$$

$$\approx \frac{1}{\sigma^2}\bar{\boldsymbol{T}}^T \hat{\boldsymbol{\epsilon}}\hat{\boldsymbol{\alpha}}^T$$

$$\nabla_{\boldsymbol{W}} \ln P_{\boldsymbol{\theta}}(\mathbf{I}) \approx \mathbb{E}_{\mathbf{s} \sim P_{\boldsymbol{\theta}}(\mathbf{s}|\mathbf{I}, \hat{\boldsymbol{\alpha}})} [\frac{1}{\sigma^2} (\hat{\boldsymbol{\epsilon}}(\boldsymbol{T}(\mathbf{s})\boldsymbol{\Phi}\hat{\boldsymbol{\alpha}})^T + \boldsymbol{\Phi}\hat{\boldsymbol{\alpha}}(\boldsymbol{T}(\mathbf{s})^T\hat{\boldsymbol{\epsilon}})^T)\boldsymbol{W}]$$

$$\approx \frac{1}{\sigma^2}(\bar{\hat{\boldsymbol{\epsilon}}}(\bar{\boldsymbol{T}}\boldsymbol{\Phi}\hat{\boldsymbol{\alpha}})^T + \boldsymbol{\Phi}\hat{\boldsymbol{\alpha}}(\bar{\boldsymbol{T}}^T\bar{\hat{\boldsymbol{\epsilon}}})^T)\boldsymbol{W}$$

where $\bar{\boldsymbol{T}} = \mathbb{E}_{\mathbf{s} \sim P_{\boldsymbol{\theta}}(\mathbf{s}|\mathbf{I}, \hat{\boldsymbol{\alpha}})}[\boldsymbol{T}(\mathbf{s})]$ and $\bar{\hat{\boldsymbol{\epsilon}}} = \mathbb{E}_{\mathbf{s} \sim P_{\boldsymbol{\theta}}(\mathbf{s}|\mathbf{I}, \hat{\boldsymbol{\alpha}})}[\hat{\epsilon}]$.

We found by chance that the approximate gradient works better than the exact gradient in practice. However, we currently do not have a theory for why the approximate gradient works better.

## Appendix F. Usage of FISTA in LSC

FISTA is a method for fast gradient descent when the objective function is a sum of a smooth convex function $f$ and a non-smooth convex function $g$ (Beck and Teboulle, 2009). It is typically applied to problems such as the traditional sparse coding, where the objective is the sum of the smooth convex function $||\mathbf{I} - \boldsymbol{\Phi}\boldsymbol{\alpha}||_2^2$ and non-smooth convex sparsity cost $||\boldsymbol{\alpha}||_1$. As an extension of sparse coding, we would like to use FISTA in order to speed up convergence for $\boldsymbol{\alpha}$ as well. The main problem is that our new objective is possibly a non-convex function of $\boldsymbol{\alpha}$, and as a result the theoretical guarantees of FISTA may not apply. Fortunately, we find that despite the lack of theoretical guarantees, FISTA still works very well in LSC.

In LSC, we directly applied FISTA with constant step size to perform the optimization problem

$$\arg \min_{\boldsymbol{\alpha}} - \ln P_{\boldsymbol{\theta}}(\boldsymbol{\alpha}|\mathbf{I}) = \arg \max_{\boldsymbol{\alpha}}(-\ln P_{\boldsymbol{\theta}}(\mathbf{I}|\boldsymbol{\alpha}) - \ln P(\boldsymbol{\alpha})) \equiv \arg \max_{\boldsymbol{\alpha}}(f(\boldsymbol{\alpha}) + g(\boldsymbol{\alpha}))$$

where $g$ corresponds to the non-smooth convex function assumed in the FISTA paper. The only free parameter is the choice of step size, which, accoridng to FISTA, should be set as $1/L(f)$ if $f(\boldsymbol{\alpha})$ were convex, where $L(f)$ is a Lipschitz constant of $f$. In our case, we set the step size as $1.5||\frac{1}{\sigma^2}\boldsymbol{\Phi}^T WW^T \boldsymbol{\Phi}||$, where $|| \cdot ||$ is the spectral norm of the matrix. To understand this choice of step size, we begin by noting that

$$\nabla_{\boldsymbol{\alpha}} f(\boldsymbol{\alpha}) = -\nabla_{\boldsymbol{\alpha}} \ln P_{\boldsymbol{\theta}}(\mathbf{I}|\boldsymbol{\alpha})$$

$$= \mathbb{E}_{\mathbf{s} \sim P_{\boldsymbol{\theta}}(\mathbf{s}|\mathbf{I}, \boldsymbol{\alpha})}[-\nabla_{\boldsymbol{\alpha}} \ln P_{\boldsymbol{\theta}}(\mathbf{I}|\mathbf{s}, \boldsymbol{\alpha})]$$

$$= \int_{\mathbf{s}} P_{\boldsymbol{\theta}}(\mathbf{s}|\mathbf{I}, \boldsymbol{\alpha})\boldsymbol{h}(\mathbf{s}, \boldsymbol{\alpha})$$

where $h(\mathbf{s}, \boldsymbol{\alpha}) \equiv -\nabla_{\boldsymbol{\alpha}} \ln P_{\boldsymbol{\theta}}(\mathbf{I}|\mathbf{s}, \boldsymbol{\alpha}) = -\frac{1}{\sigma^2}\boldsymbol{\Phi}^T\boldsymbol{T}(\mathbf{s})^T(\mathbf{I} - \boldsymbol{T}(\mathbf{s})\boldsymbol{\Phi}\boldsymbol{\alpha})$ is the score. Hence

$$\nabla_{\boldsymbol{\alpha}}^2 f(\boldsymbol{\alpha}) = \nabla_{\boldsymbol{\alpha}} \int_{\mathbf{s}} P_{\boldsymbol{\theta}}(\mathbf{s}|\mathbf{I}, \boldsymbol{\alpha})h(\mathbf{s}, \boldsymbol{\alpha})$$

$$= \int_{\mathbf{s}} h(\mathbf{s}, \boldsymbol{\alpha})\nabla_{\boldsymbol{\alpha}} P_{\boldsymbol{\theta}}(\mathbf{s}|\mathbf{I}, \boldsymbol{\alpha})^T + P_{\boldsymbol{\theta}}(\mathbf{s}|\mathbf{I}, \boldsymbol{\alpha})\nabla_{\boldsymbol{\alpha}} h(\mathbf{s}, \boldsymbol{\alpha})$$

$$= \int_{\mathbf{s}} P_{\boldsymbol{\theta}}(\mathbf{s}|\mathbf{I}, \boldsymbol{\alpha})[h(\mathbf{s}, \boldsymbol{\alpha})\nabla_{\boldsymbol{\alpha}} \ln P_{\boldsymbol{\theta}}(\mathbf{s}|\mathbf{I}, \boldsymbol{\alpha})^T + \nabla_{\boldsymbol{\alpha}} h(\mathbf{s}, \boldsymbol{\alpha})]$$

$$= \int_{\mathbf{s}} P_{\boldsymbol{\theta}}(\mathbf{s}|\mathbf{I}, \boldsymbol{\alpha})h(\mathbf{s}, \boldsymbol{\alpha})(\nabla_{\boldsymbol{\alpha}} \ln P_{\boldsymbol{\theta}}(\mathbf{I}|\mathbf{s}, \boldsymbol{\alpha}) - \nabla_{\boldsymbol{\alpha}} \ln P_{\boldsymbol{\theta}}(\mathbf{I}|\boldsymbol{\alpha}))^T$$

$$+ \int_s P_{\boldsymbol{\theta}}(\mathbf{s}|\mathbf{I}, \boldsymbol{\alpha})\frac{1}{\sigma^2}\boldsymbol{\Phi}^T W W^T \boldsymbol{\Phi}$$

$$= \frac{1}{\sigma^2}\boldsymbol{\Phi}^T W W^T \boldsymbol{\Phi} - \mathbb{E}_{s \sim P_{\boldsymbol{\theta}}(\mathbf{s}|\mathbf{I}, \boldsymbol{\alpha})}[h(\mathbf{s}, \boldsymbol{\alpha})h(\mathbf{s}, \boldsymbol{\alpha})^T] + \nabla_{\boldsymbol{\alpha}} \ln P_{\boldsymbol{\theta}}(\mathbf{I}|\boldsymbol{\alpha})\nabla_{\boldsymbol{\alpha}} \ln P_{\boldsymbol{\theta}}(\mathbf{I}|\boldsymbol{\alpha})^T$$

$$= \frac{1}{\sigma^2}\boldsymbol{\Phi}^T W W^T \boldsymbol{\Phi} - \mathbb{E}_{s \sim P_{\boldsymbol{\theta}}(\mathbf{s}|\mathbf{I}, \boldsymbol{\alpha})}[h(\mathbf{s}, \boldsymbol{\alpha})h(\mathbf{s}, \boldsymbol{\alpha})^T]$$

$$+ \mathbb{E}_{s \sim P_{\boldsymbol{\theta}}(\mathbf{s}|\mathbf{I}, \boldsymbol{\alpha})}[h(\mathbf{s}, \boldsymbol{\alpha})]\mathbb{E}_{s \sim P_{\boldsymbol{\theta}}(\mathbf{s}|\mathbf{I}, \boldsymbol{\alpha})}[h(\mathbf{s}, \boldsymbol{\alpha})]^T$$

$$= \frac{1}{\sigma^2}\boldsymbol{\Phi}^T W W^T \boldsymbol{\Phi} - \text{Cov}(h(\mathbf{s}, \boldsymbol{\alpha}))$$

Since $\boldsymbol{\Phi}^T W W^T \boldsymbol{\Phi}$ is positive semidefinite and $-\text{Cov}(h(\mathbf{s}, \boldsymbol{\alpha}))$ is negative semidefinite, the sum is not guaranteed to be positive semidefinite. Hence $f(\boldsymbol{\alpha})$ is not necessarily convex. However, it does imply that

$$||\nabla_{\boldsymbol{\alpha}}^2 f(\boldsymbol{\alpha})|| \leq ||\frac{1}{\sigma^2}\boldsymbol{\Phi}^T W W^T \boldsymbol{\Phi}|| + || - \text{Cov}(h(\mathbf{s}, \boldsymbol{\alpha}))|| \leq ||\frac{1}{\sigma^2}\boldsymbol{\Phi}^T W W^T \boldsymbol{\Phi}|| + ||\text{Cov}(h(\mathbf{s}, \boldsymbol{\alpha}))||$$

If $f(\boldsymbol{\alpha})$ were convex, then this would mean that a Lipschitz constant for $f$ would be $||\frac{1}{\sigma^2}\boldsymbol{\Phi}^T W W^T \boldsymbol{\Phi}|| + M$, where $M$ is a bound for $||\text{Cov}(h(\mathbf{s}, \boldsymbol{\alpha}))||$. $M$ is difficult to compute, but in practice we find that setting the step size as $1.5||\frac{1}{\sigma^2}\boldsymbol{\Phi}^T W W^T \boldsymbol{\Phi}||$ works well, which suggests $M$ is usually small in comparison to the first term. This is not surprising, especially since during the later stages of training the variance of $\mathbf{s}$ is usually very small, and hence the covariance of $h(\mathbf{s}, \boldsymbol{\alpha})$ is also expected to have a small spectral norm.

## Appendix G. Training datasets

Example images from the 2D translation and the rotation + scaling datasets are shown in Figure 4.

## Appendix H. More examples of image reconstruction and latent traversals

In Figure 5 and Figure 6 we provide more examples of the image reconstructions and latent traversals shown in Figure 2 and Figure 3 respectively.

## Appendix I. Training details

For training on the 2D translation and the rotation + scaling dataset, the hyperparameters used for the model is detailed in Table 2.

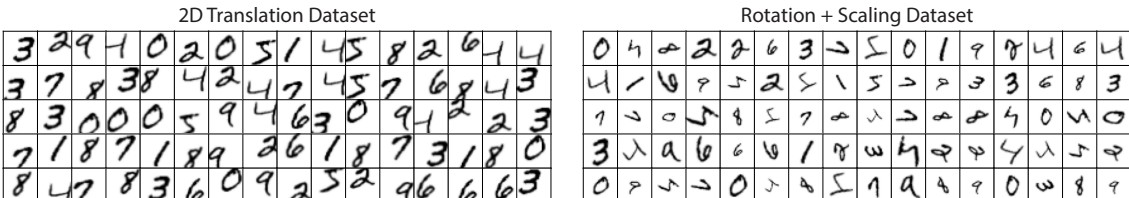

Figure 4: 80 example images from each of the two synthetic datasets

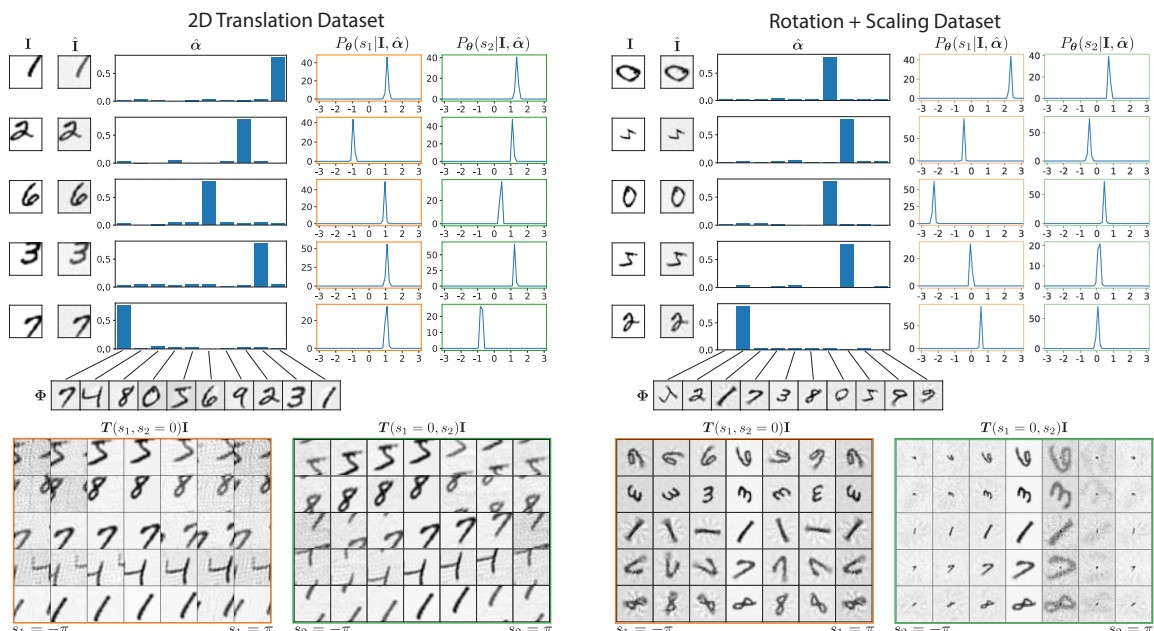

Figure 5: 80 example images from each of the two synthetic datasets

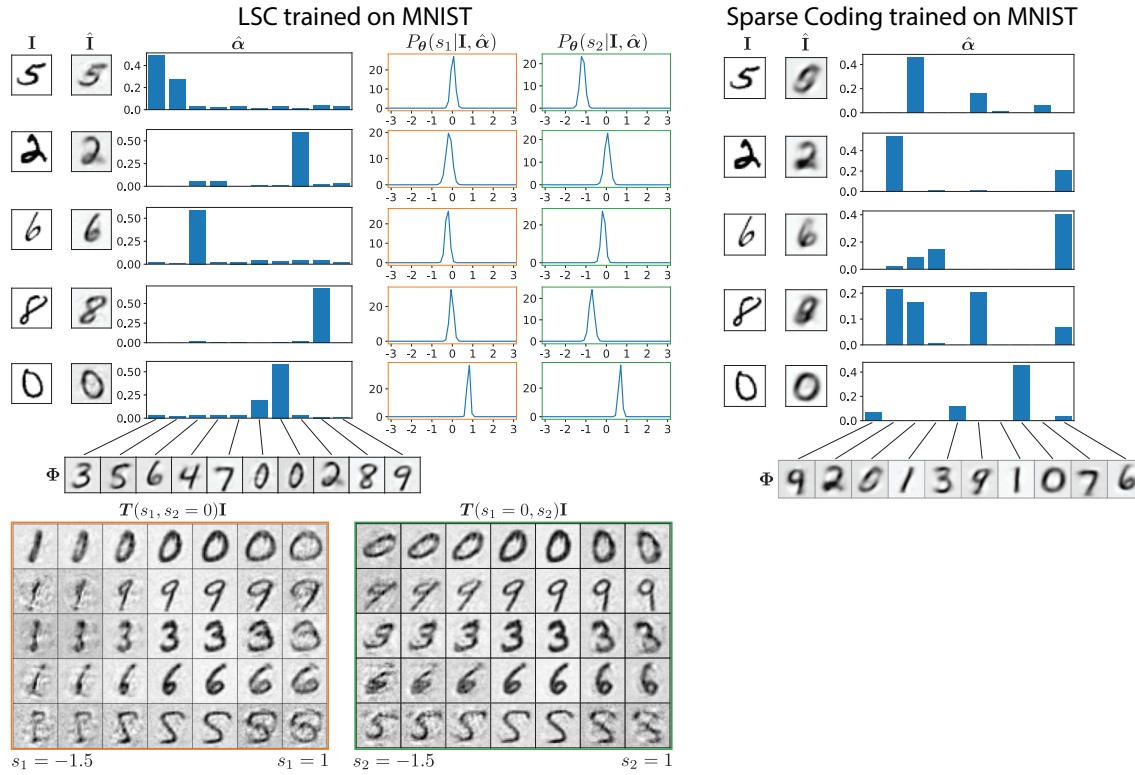

Figure 6: 80 example images from each of the two synthetic datasets

Table 2: Hyperparameters of LSC when trained on 2D translation and rotation + scaling datasets

| Variable | Value |
| --- | --- |
| $B$ (batch size) | 100 |
| $K$ (number of dictionary templates) | 10 |
| $N$ (number of samples along each dim of the integral $\bar{\boldsymbol{R}} = \int_{\mathbf{s}} P_{\boldsymbol{\theta}}(\mathbf{s}|\mathbf{I}, \boldsymbol{\alpha})\boldsymbol{R}(\mathbf{s}))$ | 50 |
| $L$ (number of irreducible representations) | 128 |
| $T$ (number of gradient update steps for $\boldsymbol{\alpha}$) | 20 |
| $n$ (dimensionality of transformation parameter $\mathbf{s}$) | 2 |
| $\sigma^2$ (variance of the Gaussian noise $\epsilon$ in the generative model) | 0.01 |
| $\lambda$ (sparse penalty) | 10 |
| $\eta_{\boldsymbol{\Phi}}$ (learning rate for $\boldsymbol{\Phi}$) | 0.05 |
| $\eta_{\boldsymbol{W}}$ (learning rate for $\boldsymbol{W}$) | 0.3 |
| $\boldsymbol{\alpha}_0$ (initialization of $\boldsymbol{\alpha}$) | 0.01 |
| multiplicity of $\boldsymbol{\omega}$ | 1 |
| parameters for geoopt Riemannian ADAM optimizer (excluding learning rate) | default |

When trained on MNIST, the only change is that the multiplicity of $\boldsymbol{\omega}$ is 2 instead of 1.

For computing the SNRs in Table 1, we used $N = 100$ instead of $N = 50$ to obtain higher quality reconstruction. Moreover, a warmup period is applied to the learning rates of both $\boldsymbol{\alpha}$ and $\boldsymbol{W}$, where the learning rates are linearly increased from 0 to the specified values in Table 2 over 3 epochs. This greatly reduces the trial-to-trial fluctuation of image reconstruction quality, and its usage is highly recommended as there are no observed drawbacks. The SNRs for sparse coding are obtained using an improved version of the algorithm in Olshausen and Field (1997), where the main modification is the use of FISTA in optimizing $\boldsymbol{\alpha}$. To ensure fair comparison between LSC and sparse coding, the following hyperparameters for both models are set to be equal: B, K, $\sigma^2$, and $\lambda$. Notice that when these hyperparameters are set equal, the loss function for LSC coincides with the loss function for sparse coding in the limiting case where all the $\boldsymbol{\omega}_l$ are 0 (in which case the transformation $\boldsymbol{T}(\mathbf{s})$ will just be the identity matrix) and $\boldsymbol{W}$ is full rank. The learning rate for $\boldsymbol{\Phi}$ in sparse coding algorithm is manually optimized for best possible SNRs, and we find $\eta_{\boldsymbol{\Phi}} = 0.05$ works best.

All training is done on a Nvidia GTX 1080Ti GPU.

