# OpenReview forum: "Disentangling Images with Lie Group Transformations and Sparse Coding"
_NeurIPS.cc/2022/Workshop/NeurReps — NeurReps 2022 Poster_

### Official Review · Reviewer_89Pm · 2022-10-11
**Nice work!**

**Confidence:** 4
**Soundness:** 3
**Presentation:** 4
**Contribution:** 3
**Overall Rating:** 6

**Summary:**

This paper descirbes a spase coding model with an additional component that can learn transformations of the latents. The transformations are constrained via Lie groups. The model learns a reduced set of sparse bases since the transformation can account for roations etc.

**Questions:**

My only real issue is the use of the word disentanling. It is not really disentangling in the normal use of the work as there is an architectural bias where one model component can only learn transformations, and the other can only learn bases. But all fine by me.

**Limitations:**

Currently it's a nice proof of principle, but to really get people excited the ideas need to be shown to be useful over other methods in some domain.

**Recommended Decision:**

3: Accept

**Relevance:**

4: Highly relevant

**Strengths And Weaknesses:**

Originality: This is a nice extension of the ideas of Gklezakos & Rao, 2017 so that the types of transformations are not predefined.
Quality: I have no concerns.
Clarity: Very good.
Significance: The results will be of interest to the community.

**Submission Track:**

Proceedings Paper (9 Page)

---

### Official Review · Reviewer_FEJW · 2022-10-14
**A nice little bayesian algorithm combining sparse coding and continuous transformations for images**

**Confidence:** 4
**Soundness:** 4
**Presentation:** 4
**Contribution:** 3
**Overall Rating:** 7

**Summary:**

In this work the authors present a novel bayesian algorithm for unsupervised learning on images. It combines two approaches: sparse coding and Lie group transformation, both of which have been studied previously in the literature, to which the authors point in their excellent literature review. They combine them and show a method for training the model on a set of images. They test their model on datasets made from subsets of MNIST transformed in only one or two ways and show the model successfully learns to disentangle object ID from transformation. Then they train it on the full MNIST dataset and show it performs quite well (but not perfectly) at disentangling image ID from some squishing and bending factors. This is a nice proof of concept.

**Questions:**

I just have a couple of questions:

1) What does the multiplicity of frequencies allow? The same frequency may be involved in multiple different transformation types?

2) Do the authors agree that in an ideal world you would be able to learn the positions of the frequencies $\omega_l$? If so is there just no nice probabilistic framework for doing that?

3) If I have understood figure 2 correctly then for rotation they should be radially symmetric oscillations at different frequencies? And scaling makes them vary with radius? Further, the scaling group is an example of a group that is not really the torus, but somehow your scheme is still able to learn?


**Limitations:**

The authors fairly discussed their limitations,

**Recommended Decision:**

3: Accept

**Relevance:**

4: Highly relevant

**Strengths And Weaknesses:**

Originality

The work seemed clearly original. It was a combination of two existing ideas, but they were combined carefully.

Quality

The submission seemed very sound, and the claims were well supported

Clarity

I thought the paper was very clearly presented and nice to read. There was a tendency towards a lot of stuff, including the literature review, which had to have a paragraph shoved in the appendix, and I was surprised by the amount of appendices dedicated to deriving the real irreps of $\mathbb{T}^n$, is this not well known stuff? But the main paper was largely neatly argued and presented.

Significance

I don't feel very well placed to judge this, but I enjoyed it and it seemed interesting.

**Submission Track:**

Proceedings Paper (9 Page)

---

### Official Review · Reviewer_zUe2 · 2022-10-16
**Useful extension of sparse coding to incorporate common image transformations**

**Confidence:** 3
**Soundness:** 3
**Presentation:** 4
**Contribution:** 3
**Overall Rating:** 7

**Summary:**

The authors extend the original sparse coding model of Olshausen and Field to incorporate a subclass of common image transformations including rotations, translations, and scalings. This was done by parameterizing these transformations using their linear representations as elements of the corresponding compact, connected, commutative Lie groups. This added orthonormal transformation basis parameter $W$ and transformation latents $s$ to the dictionary parameter $\Phi$ and decomposition latents $\alpha$ of the original sparse coding model. The parameters $\theta = (W, \Phi)$ were learned using variational EM by approximating the posterior on latents as $p_\theta(s, \alpha | I) = p_\theta(\alpha|I) p_\theta(s|\alpha,I) \approx \delta(\alpha - \hat \alpha) p_\theta(s|I, \hat \alpha)$, where the MAP estimate $\hat \alpha$ was computed by FISTA during the E-step, and the parameters were updated by gradient descent and normalization during the M-step. Experiments on transformed MNIST digits showed that the model was able to distentangle the shapes of the digits from the transformations applied to them, which the original sparse coding model with matched dictionary size was unable to do.

**Questions:**

1. How was the transformation basis W initialized and how robust was the model to these initializations i.e. did most initializations converge to good representations?
2. Both the original sparse coding model and the current extension have hyperparameters like the number of dictionary elements K, or the sparsity parameter $\lambda$. In their experiments the authors fixed these at appropriate values, but in practice they must be learned from the data. Do the authors anticipate any additional difficulties in learning these parameters for their proposed model relative to the original sparse coding model?
3. Lie group ideas motivated the parameterization the authors used to extend the original sparse coding model, but there was nothing in their optimization procedure that was particular to the parameterization. This suggests that their approach can be applied to a wider range of parameterizations, potentially allowing a larger range of transformations to be disentangled. For example, the condition that the 2D blocks of $R(s)$ be rotations can be relaxed to allow these blocks to be arbitrary, potentially expanding the range of transformations the model can learn. Can the authors comment on whether their model can be fruitfully extended along these lines?

**Limitations:**

The authors did a good job of mentioning the limitations of their work, both with regards to the transformations that can be learned, and the computational costs of the optimization e.g. the integrations over $s$ required to update the parameters. The authors should perhaps provide more information about the time course of learning, robustness to initialization, and the learning of the model hyperparameters.

**Recommended Decision:**

3: Accept

**Relevance:**

3: Solid fit

**Strengths And Weaknesses:**

The submission was clearly written, well contexualized, and technically sound (as far as the main text goes - I didn't read all of the appendices in detail). The work is a simple combination of two existing ideas (sparse coding, and linear representations of Lie groups), so while not particularly original, is a demonstration that such a combination can be practically carried out to yield a useful and interesting result, namely the disentanglement of shapes from the transformations that are applied to them. I believe this is an important result that will be of interest to the community.

**Submission Track:**

Proceedings Paper (9 Page)

---

### Decision · Program_Chairs · 2022-10-21

Accept (Poster)